# Ubiquitin and TFIIH-stimulated DDB2 dissociation drives DNA damage handover in nucleotide excision repair

Cristina Ribeiro-Silva[1], Mariangela Sabatella[1,2], Angela Helfricht[1], Jurgen A. Marteijn[1], Arjan F. Theil[1], Wim Vermeulen[1✉] & Hannes Lans[1✉]

DNA damage sensors DDB2 and XPC initiate global genome nucleotide excision repair (NER) to protect DNA from mutagenesis caused by helix-distorting lesions. XPC recognizes helical distortions by binding to unpaired ssDNA opposite DNA lesions. DDB2 binds to UV-induced lesions directly and facilitates efficient recognition by XPC. We show that not only lesion-binding but also timely DDB2 dissociation is required for DNA damage handover to XPC and swift progression of the multistep repair reaction. DNA-binding-induced DDB2 ubiquitylation and ensuing degradation regulate its homeostasis to prevent excessive lesion (re)binding. Additionally, damage handover from DDB2 to XPC coincides with the arrival of the TFIIH complex, which further promotes DDB2 dissociation and formation of a stable XPC-TFIIH damage verification complex. Our results reveal a reciprocal coordination between DNA damage recognition and verification within NER and illustrate that timely repair factor dissociation is vital for correct spatiotemporal control of a multistep repair process.

[1] Department of Molecular Genetics, Oncode Institute, Erasmus MC, University Medical Center Rotterdam, Dr. Molewaterplein 40, 3015 GD Rotterdam, The Netherlands. [2] Present address: Princess Máxima Center for pediatric oncology, Heidelberglaan 25, 3584 CS Utrecht, The Netherlands. ✉email: w.vermeulen@erasmusmc.nl; w.lans@erasmusmc.nl

Global-genome nucleotide excision repair (GG-NER) is an essential DNA repair machinery that protects cells against a wide range of structurally unrelated DNA lesions, including the highly mutagenic UV-induced cyclobutane-pyrimidine dimers (CPDs) and 6-4 pyrimidine-pyrimidone photoproducts (6-4PPs)[1–3]. If not repaired, these lesions interfere with transcription and replication, thereby compromising genomic stability and instigating mutagenesis associated with premature aging and skin cancer[4,5]. In mammalian cells, GG-NER is initiated by the main damage sensor XPC, as part of the heterotrimeric XPC-CETN2-RAD23B complex, whose substrate versatility derives from its indirect damage recognition mode[6]. As XPC diffuses through the nucleus, it continuously probes DNA searching for thermodynamically helix-destabilized structures[7] that allow the intercalation of its double β-hairpin domain into the DNA before dissociation[8–10]. In this way, XPC captures and binds extruding nucleotides in the undamaged strand without contacting the lesion itself[11].

XPC recruitment to UV-induced DNA damage is stimulated by the UV-DDB complex, comprising of DDB1 and DDB2[6,12]. DDB2 binds directly to and flips out UV-induced damaged bases to create a more suitable substrate for XPC[12–16]. This activity is particularly relevant for GG-NER of CPDs, which generate only minor DNA helix distortions that are, otherwise, not efficiently recognized by XPC[17]. In addition, DDB2 is thought to facilitate XPC recruitment within chromatinized DNA through its ability to promote chromatin reorganization[18,19]. The UV-DDB complex is part of a larger E3 ubiquitin-ligase complex (CRL4$^{DDB2}$), also containing CUL4A, RBX1, and the COP9 signalosome[20]. When DDB2 binds to UV-lesions the COP9 signalosome dissociates, which stimulates the E3 ubiquitin-ligase activity of the complex[20,21]. Several proteins were reported to be ubiquitylated by CRL4$^{DDB2}$, including core histones H2A, H3 and H4, XPC and DDB2 itself[20,22–25].

Because XPC also detects mismatches and other DNA helix distortions that are not processed by nucleotide excision repair (NER), subsequent damage verification plays a crucial role in ensuring the fidelity of NER. XPC binding to helix-destabilizing lesions recruits the transcription factor IIH (TFIIH) complex through interactions with its helicase XPB and core GTF2H1 (also known as p62) subunits[26–28]. TFIIH's other helicase, XPD, verifies the presence of genuine NER substrates by unwinding the DNA in 5′–3′ direction while scanning for helicase blocking lesions[29,30]. Damage verification is stimulated by the DNA damage binding protein XPA, which, together with the ssDNA binding RPA protein complex, also recruits and positions the endonucleases XPF-ERCC1 and XPG, completing the formation of the pre-incision complex. DNA incision 5′ and 3′ of the lesion by XPF-ERCC1 and XPG, respectively, leads to the removal of a 22–30 nucleotide long ssDNA enclosing the lesion[2,3,31]. The resulting gap is restored by de novo DNA synthesis and ligation[32].

Due to the complexity of the dynamic arrangement of NER factors, temporal and spatial coordination of each NER step is required for efficient repair and accurate restoration of damaged DNA. The sequential damage detection, verification, excision and gap-filling steps give NER the appearance of a linearly ordered, multistep cascade. However, how the progression from one step to the next is coordinated and how each of these consecutive steps feed back onto each other is not yet fully known. The early steps of GG-NER are under tight control by post-translational protein modifications (PTMs), likely to ensure proper damage handover to subsequent NER steps. For instance, the CRL4$^{DDB2}$ complex catalyzes the polyubiquitylation of DDB2 after binding to UV lesions, as well as monoubiquitylation of histone H2A[25], stimulating DDB2 extraction

from DNA by the ubiquitin-dependent segregase p97/VCP and targeting it for proteasomal degradation[21,33–35]. Furthermore, CRL4$^{DDB2}$ reversibly ubiquitylates XPC, which was suggested to stabilize its association with DNA[23]. Subsequent sumoylation[36–38] and RNF111-mediated[39] ubiquitylation of XPC were suggested to promote its dissociation to favor XPG binding. Besides, Poly [ADP-ribose] polymerase 1 activity appears to fine-tune the E3 ubiquitin-ligase activity of the CRL4$^{DDB2}$ complex and the ubiquitylation and DNA damage binding of XPC[40] and DDB2[41,42]. Despite extensive evidence of PTM-mediated regulation of both DDB2 and XPC, it is still unclear how, once the damage is detected, the DNA association and dissociation of XPC and DDB2, respectively, are coordinated with the recruitment of TFIIH to execute damage verification.

In this study, we show that damage verification differently feeds back on DDB2 and XPC, as TFIIH recruitment coincides with DDB2 dissociation but stabilizes XPC binding to damaged chromatin. Interestingly, although binding of DDB2 to DNA damage is required for optimal repair of UV-induced lesions, its timely dissociation after damage detection is needed to promote the formation of a stable XPC-TFIIH-DNA complex. Our results suggest that the ubiquitylation and proteolytic degradation of DDB2 regulate its DNA damage sensing activity by limiting its availability, thus facilitating proper damage handover and the swift progress of the NER reaction.

## Results

**DDB2 and XPC are differently regulated by downstream factors.** We studied how, in living cells, the association of DDB2 and XPC with DNA damage is affected by the recruitment of the downstream NER machinery that verifies and excises the damage. To this end, we measured the UV-C induced change in mobility of GFP-tagged DDB2 and XPC with fluorescence recovery after photobleaching (FRAP). Incomplete fluorescence recovery reflects transient immobilization of GFP-tagged proteins, such as binding to damaged DNA[7,43,44]. A change in the immobile fraction after UV, therefore, indicates that either less or more proteins are bound to damaged DNA or that each protein is bound for a shorter or longer time. SV40-immortalized human fibroblasts stably expressing GFP-DDB2 or XPC-GFP were treated with siRNA against either GTF2H1, to interfere with damage verification, or against XPG, to block excision, or with non-targeting siRNA as control (CTRL) (knockdown efficiencies of siRNAs used are shown in Supplementary Fig. 1). Following UV-irradiation, a significant fraction of DDB2 molecules was transiently bound to UV-damaged DNA (Fig. 1a, b). Interestingly, this UV-induced DDB2 immobilization increased after the depletion of GTF2H1 and, to a lesser extent, also after XPG knockdown (Fig. 1a, b). Also, UV-induced XPC immobilization increased after XPG knockdown. In striking contrast, however, XPC binding decreased when GTF2H1 was depleted (Fig. 1c, d; Supplementary Fig. 1i, j). These observations show that downstream NER proteins differentially regulate DDB2 and XPC. While damage verification via TFIIH promotes stable XPC binding to damaged DNA, it appears that TFIIH recruitment coincides with or even stimulates DDB2 dissociation, possibly to allow proper damage verification. However, when the verification step is still intact but the excision of DNA damage is blocked (i.e., with siXPG), the binding of both DDB2 and XPC to damaged DNA increases. The slowly ascending slopes of the FRAP curves after UV (Fig. 1a, c) suggest that both DDB2 and XPC molecules are not statically bound but are also released within the time course of the FRAP experiments, reflecting dynamic binding and dissociation.

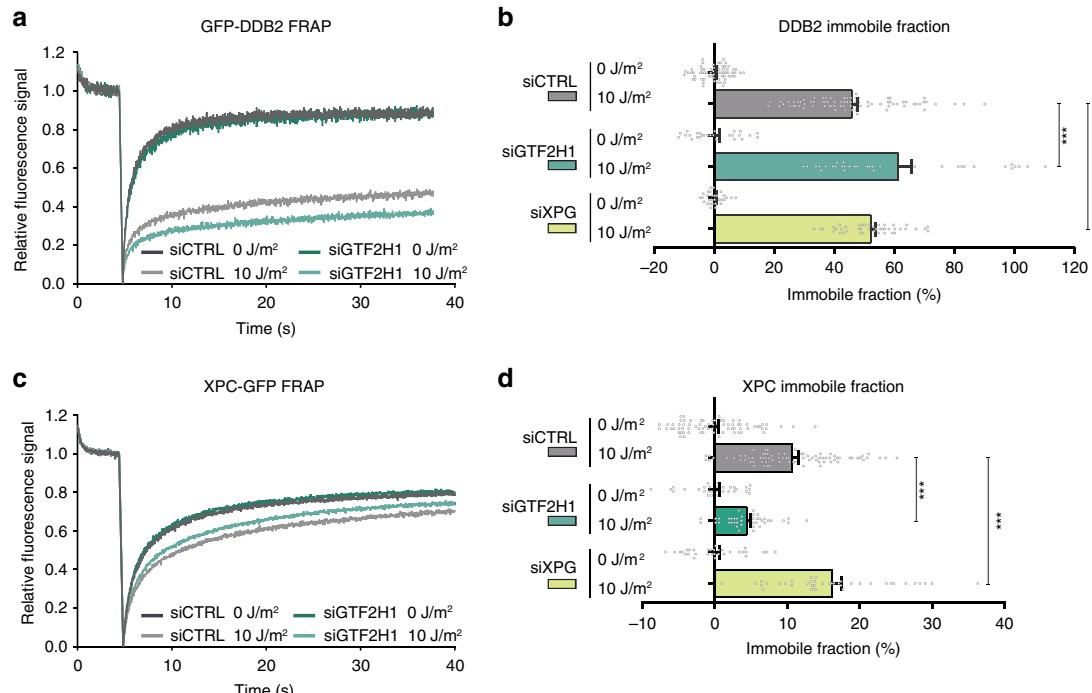

**Fig. 1 DDB2 and XPC are differently regulated by downstream factors. a** Fluorescence Recovery After Photobleaching (FRAP) analysis of DDB2 mobility in mock or UV-C irradiated (10 J/m$^2$) VH10 cells stably expressing GFP-DDB2 and transfected with control (CTRL) or GTF2H1 siRNAs. GFP-DDB2 fluorescence recovery was measured in a strip across the nucleus after bleaching, normalized to bleach depth, and the average pre-bleach intensities (1.0). **b** Percentage of GFP-DDB2 immobile fraction in VH10 fibroblasts treated with control (CTRL), GTF2H1 or XPG siRNAs, determined from FRAP analyses as depicted in (**a**). Percentage immobile fraction represents the ratio between the average recovered fluorescence intensity of UV- and mock-treated cells, over the last 10 s of the measurements, as explained in the methods. **c** FRAP analysis of XPC mobility in mock or UV-C irradiated (10 J/m$^2$) XP4PA cells stably expressing XPC-GFP and transfected with control (CTRL) or GTF2H1 siRNAs. XPC-GFP-fluorescence recovery was measured and normalized as described in (**a**). **d** Percentage of XPC-GFP immobile fraction in XP4PA cells treated with control (CTRL), GTF2H1 or XPG siRNAs, determined by FRAP analysis as depicted in (**c**) and described in (**b**). Graphs and FRAP curves depict the mean & S.E.M. of >30 cells from three independent experiments. **P < 0.01, ***P < 0.001, relative to siCTRL control 10 J/m$^2$, analyzed by unpaired, two-tailed t-test (adjusted for multiple comparisons, see "Methods"). Source data are provided as a Source Data file.

**Persistent damage detection in absence of lesion excision**. To verify the increased binding of endogenous DDB2 and XPC to DNA damage in the absence of repair, we used our recently established XPF knockout (XPF KO) U2OS cells[45] as an excision-deficient model cell line in which damage verification still takes place and U2OS wild-type (WT) as a NER-proficient cell line. Using immunofluorescence (IF), we monitored the accumulation of endogenous DDB2 and XPC in time at local UV damage (LUD), generated by UV-C irradiation (60 J/m$^2$) through a microporous membrane. LUD was visualized by counterstaining for CPDs, which are only slowly repaired in human cells and, therefore, still detectable within the time course of our experiment[46].

In WT cells, DDB2 accumulated rapidly (within 10 min) at LUD and its accumulation slowly declined in time, likely reflecting the removal of easily accessible and rapidly repairable lesions (such as 6-4PPs) (Fig. 2a, b). In excision-deficient XPF KO cells, early accumulation of DDB2 did not differ greatly from that in WT cells, but at later time points (40 min, 2, and 8 h) we observed an increased accumulation of DDB2 at LUD (Fig. 2a, b). This suggests that DDB2 keeps being recruited to persisting, unrepaired lesions when these are not excised. After binding to UV-damaged DNA, DDB2 is ubiquitylated and targeted for proteasome-mediated degradation[23,33]. Thus, if DDB2 is continuously binding to and dissociating from damaged DNA, it is expected that in time, an increasing amount of DDB2 molecules would be degraded. Indeed, we noticed a significant decline in total DDB2 protein levels in

time in the locally irradiated XPF KO cells (Fig. 2c). Such decline was not observed in U2OS WT cells, apparently because the amount of DDB2 molecules that binds to LUD and is degraded is too small to be detected on the total protein level. Besides, inhibition of DDB2 degradation with proteasome inhibitor MG132 led to even higher DDB2 accumulation, persisting in time in XPF KO cells (Supplementary Fig. 2a–c). This suggests that DDB2 degradation normally prevents rebinding to lesions by downregulating its availability. In NER-proficient WT cells, however, DDB2 accumulation did not increase in the absence of proteasome activity, showing that DDB2 dissociation from damage occurs normally and is uncoupled from its subsequent degradation.

XPC also showed a rapid accumulation (within 10 min) at LUD in WT cells, which slowly diminished in time as the bulk of lesions were being removed (Fig. 2d, e). Interestingly, XPC levels did not visibly change (Supplementary Fig. 2d) and its accumulation at LUD did not decrease in time in the XPF KO cells (Fig. 2d, e). These results indicate that if lesions are not excised, the DNA damage sensing proteins DDB2 and XPC are continuously recruited to sites of DNA damage, implying that multiple rounds of damage detection keep on taking place. However, their fate after binding DNA damage is dramatically different. The accentuated DDB2 degradation could imply that the dissociation of DDB2 and its subsequent degradation are necessary for NER to proceed. XPC, on the other hand, is required for and becomes more stably bound by TFIIH recruitment.

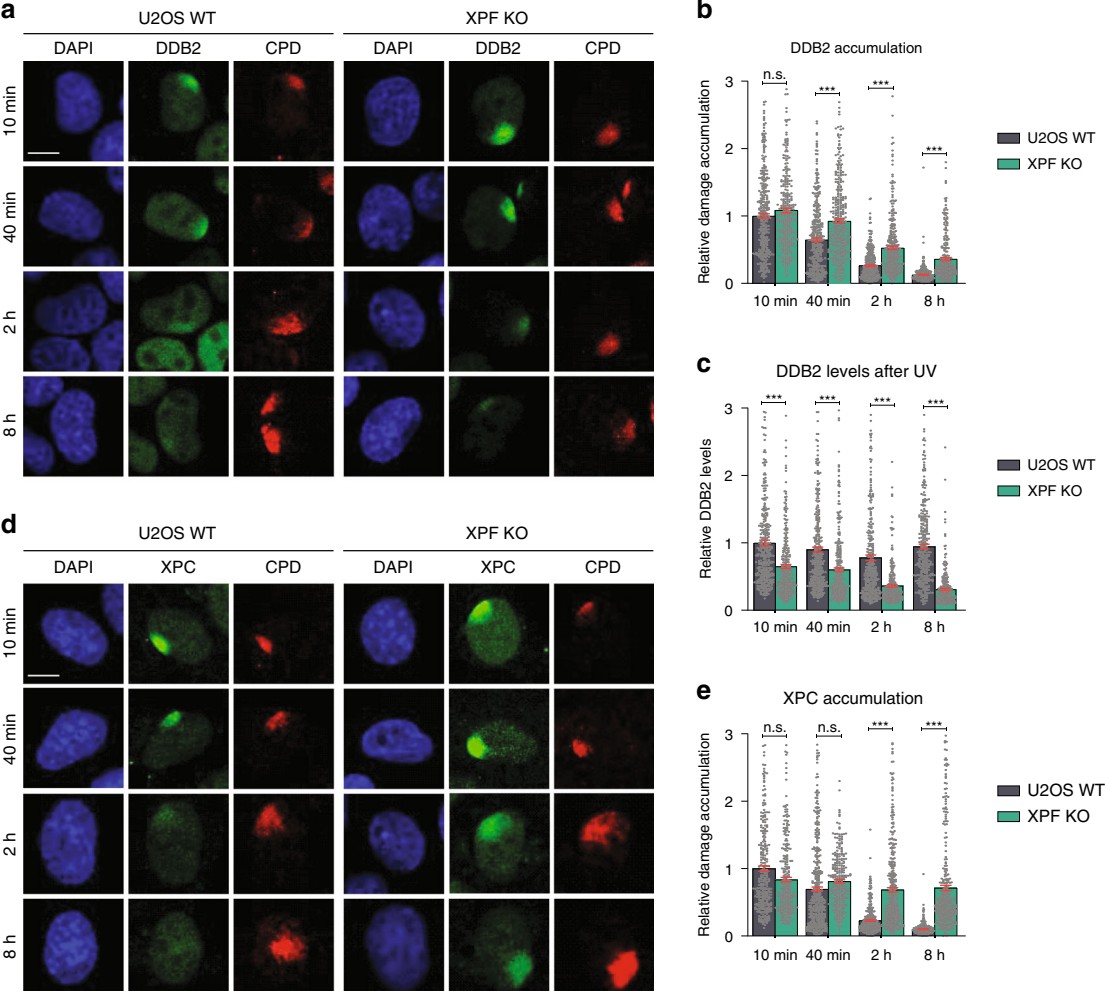

**Fig. 2 Persistent damage detection in absence of lesion excision. a** Representative immunofluorescence (IF) images of endogenous DDB2 accumulation at local UV-C damage (LUD) in U2OS wild-type (WT) and U2OS XPF knockout (XPF KO) cells. Cells were fixed 10 min, 40 min, 2 h and 8 h after LUD (marked by CPD staining) induced with UV-C irradiation ($60 \, J/m^2$) through a microporous membrane (8 μm). **b** Quantification of DDB2 accumulation at LUD, as depicted in (**a**). DDB2 accumulation was normalized to the nuclear background and U2OS WT 10 min after UV-C, which was set to 1.0. **c** Total DDB2 protein levels determined by measuring total nuclear fluorescent signal intensities in nuclei such as depicted in **a** and normalized to U2OS WT 10 min, which was set to 1.0. **d** Representative IF images of endogenous XPC accumulation at LUD in U2OS WT and XPF KO cells, as described in (**a**). **e** Quantification of XPC accumulation at LUD, as depicted in (**d**) and described in (**b**). Mean and S.E.M. of, respectively, $n = 348, 313, 383, 334, 355, 334, 316, 247$ cells for DDB2 and $n = 305, 276, 413, 272, 339, 383, 266, 339$ cells for XPC from five independent experiments. *$P < 0.05$, **$P < 0.01$, ***$P < 0.001$, analyzed by one-way ANOVA (see "Methods"). Scale bars in (**a**, **d**): 5 μm. Source data are provided as a Source Data file.

**TFIIH promotes DDB2 dissociation and stable XPC binding.** Because our FRAP analysis suggested that TFIIH recruitment promotes the stable binding of XPC to DNA damage and the dissociation of DDB2 (Fig. 1), we next tested whether endogenous DDB2 and TFIIH might exchange at sites of damaged DNA to promote efficient XPC association with damaged DNA. Using IF, we found that the depletion of GTF2H1 led to increased and prolonged accumulation of endogenous DDB2 at LUD in U2OS WT cells (Fig. 3a, b). Strikingly, even in XPF KO cells, in which DDB2 is already continuously recruited due to the complete absence of repair, depletion of GTF2H1 still led to a significantly increased and prolonged DDB2 accumulation at damage (Fig. 3a, b). This is in line with the FRAP data, showing a stronger UV-induced DDB2 immobilization after GTF2H1 than after XPG depletion (Fig. 1b) and, therefore, suggests that TFIIH recruitment coincides with, and might even promote, DDB2 dissociation. Also in support of our FRAP data, the depletion of GTF2H1 in WT cells led to a delay in XPC recruitment to LUD, i.e., XPC accumulation peaked at a later time point (Fig. 3c; Supplementary Fig. 2e).

As part of the CRL4DDB2 complex, DDB2 itself is a substrate of the complex's E3 ubiquitin-ligase activity[20,22,23,33]. Interestingly, in in vitro ubiquitylation assays, more DDB2 ubiquitylation was observed in the absence of XPC, which has led to the speculation that XPC recruitment protects DDB2 from excessive auto-ubiquitylation and degradation, thus enabling DDB2 to perform multiple rounds of damage detection[34]. As we observed increased DDB2 and delayed XPC DNA damage recruitment after GTF2H1 knockdown (Figs. 1, 3), we tested whether the absence of TFIIH at damage results in higher DDB2 ubiquitylation levels, promoting its degradation. Immunoblot analysis of GFP-DDB2 immunoprecipitated from UV-irradiated cells clearly showed a significant increase in UV-induced DDB2 ubiquitylation after siGTF2H1, marked by increased FK2 antibody staining recognizing mono- and poly-ubiquitylated protein conjugates (Fig. 3d, e; Supplementary Fig. 2f). In accordance, the depletion of GTF2H1 in U2OS cells accelerated the UV-induced and proteasome-dependent DDB2 proteolysis (Fig. 3f, g). Our observations suggest that the recruitment of TFIIH promotes the stable binding of

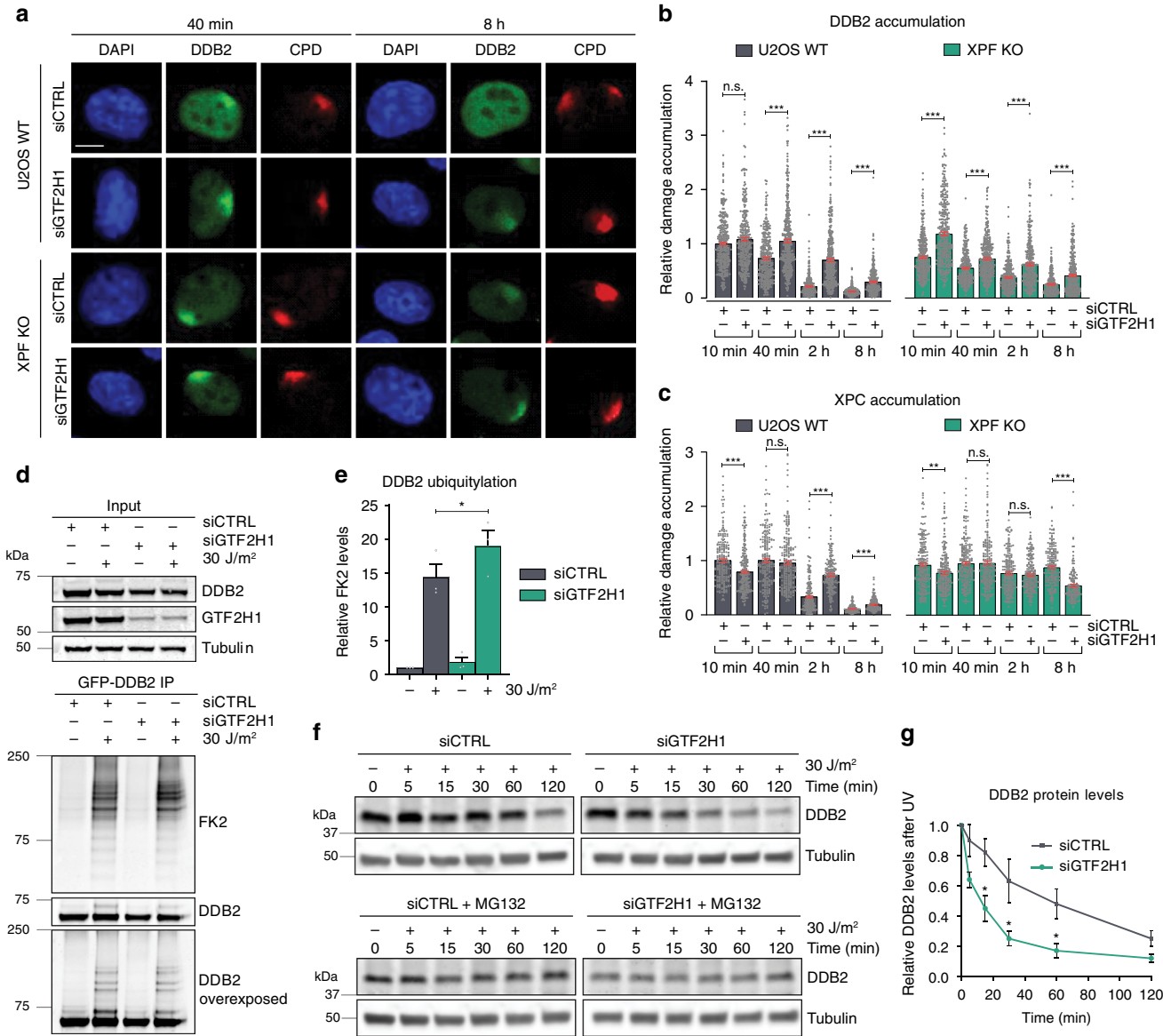

**Fig. 3 TFIIH promotes DDB2 dissociation and stable XPC binding. a** IF images of endogenous DDB2 LUD accumulation in U2OS WT and XPF KO cells treated with control (CTRL) or GTF2H1 siRNAs, 40 min and 8 h after UV-C irradiation (60 J/m$^2$) Scale bar: 5 µm. **b, c** Quantification of (**b**) DDB2, or (**c**) XPC accumulation at LUD in U2OS WT and XPF KO cells treated with CTRL or GTF2H1 siRNAs, 10 min, 40 min, 2 h and 8 h after damage, as described in (**a**), normalized to the nuclear background and U2OS siCTRL 10 min. Mean and S.E.M. of, respectively, $n = 322$, 321, 317, 374, 335, 364, 291, 299, 370, 314, 318, 342, 307, 315, 283, 287 cells in (**b**) or 217, 225, 210, 234, 220, 193, 165, 173, 218, 238, 218, 213, 217, 202, 194, 156 cells in (**c**) from two independent experiments. IF images of XPC are shown in Supplementary Fig. 2e. **d** Immunoblot showing DDB2 ubiquitylation in GFP-DDB2 VH10 cells, before or 15 min after UV-C irradiation (30 J/m$^2$) and treated with CTRL or GTF2H1 siRNAs. Total cell lysates (Input) were analyzed with DDB2, GTF2H1 and Tubulin antibodies. GFP-DDB2 immunoprecipitation (IP) fractions were analyzed using anti-ubiquitin (FK2) and DDB2 antibodies. Control IP is shown in Supplementary Fig. 2f. **e** Quantification of ubiquitin levels shown in **d**, normalized to DDB2 levels and non-irradiated siCTRL samples. Mean and S.E.M. of three independent experiments. **f** Immunoblot showing UV-induced DDB2 proteolysis in total cell lysates of U2OS cells treated with CTRL or GTF2H1 siRNAs in the absence and presence of MG132 proteasome inhibitor, at the indicated time points after UV irradiation (30 J/m$^2$) and analyzed by DDB2 and tubulin antibody. **g** Quantification of DDB2 proteolysis as depicted in (**f**), normalized to tubulin and non-irradiated samples. Mean and S.E.M. of three independent experiments. *$P < 0.05$, **$P < 0.01$, ***$P < 0.001$, n.s. non-significant, analyzed by one-way ANOVA in (**b**), (**c**) and by unpaired, two-tailed $t$-test (adjusted for multiple comparisons) in (**e**), (**g**) (see "Methods"). Source data are provided as a Source Data file.

XPC to damaged DNA and the dissociation of DDB2, thereby preventing excessive DDB2 auto-ubiquitylation and degradation.

**DDB2 retention impairs stable XPC and TFIIH damage binding.** To further investigate the interplay between TFIIH arrival and DDB2 dissociation, we devised an approach to increase the residence time of DDB2 to test whether this would affect the recruitment of XPC and TFIIH. Previously, the

ubiquitin-dependent segregase p97/VCP was shown to facilitate the extraction of ubiquitylated DDB2 from UV-damaged chromatin[35]. Therefore, we used a specific inhibitor of VCP (VCPi) to impair DDB2 chromatin extraction, and measured recruitment of DDB2 to LUD using IF (Fig. 4a, b). In the presence of VCPi, DDB2 initial accumulation at LUD was indeed higher and gradually disappeared in time, albeit with delayed kinetics (Fig. 4a, b). This was corroborated by FRAP analysis on GFP-DDB2,

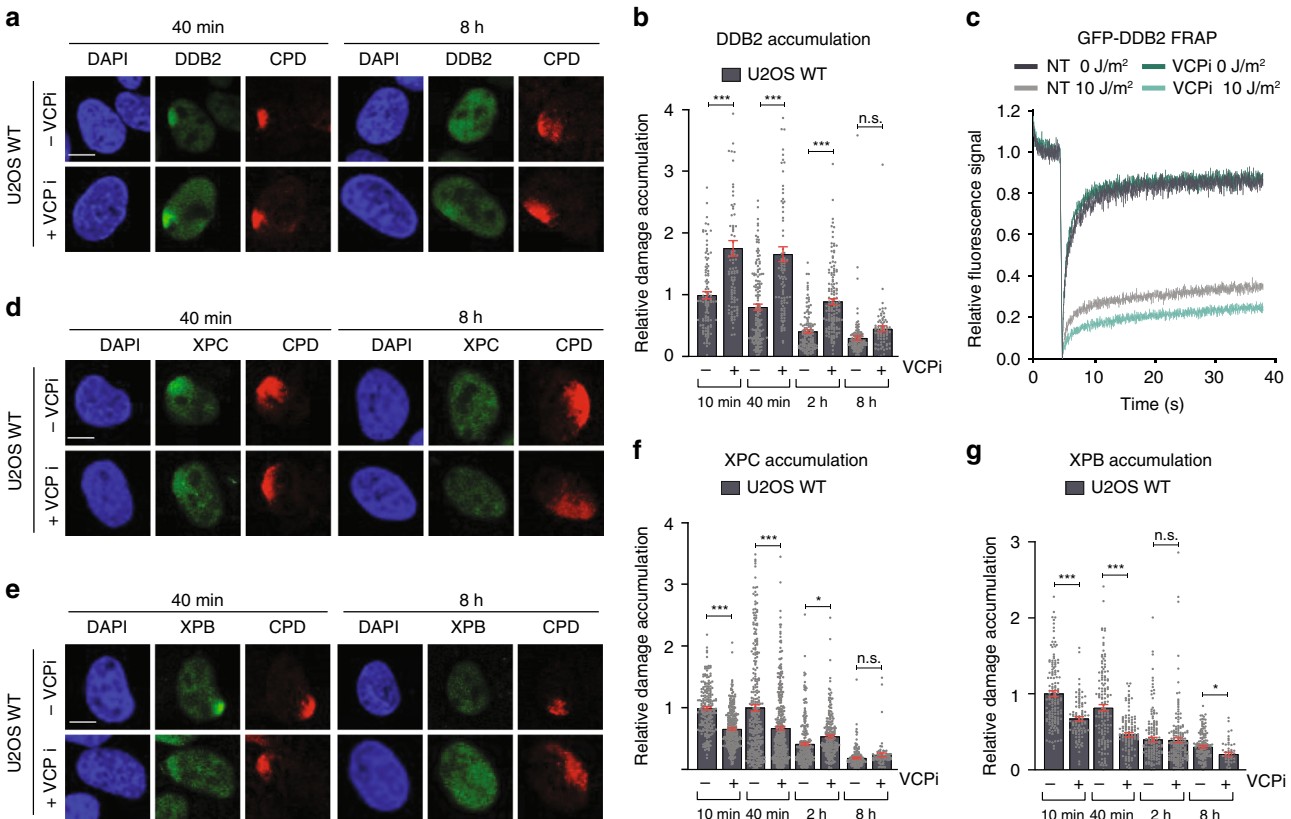

**Fig. 4 DDB2 retention impairs stable XPC and TFIIH damage binding. a** Representative IF images of endogenous DDB2 accumulation at LUD in U2OS WT cells in the absence or presence of VCP inhibitor (VCPi). 1 h before LUD induction, 10 μM VCPi was added and 40 min and 8 h after local UV irradiation (60 J/m²) through a microporous membrane (8 μm), cells were fixed and analyzed by IF. Scale bars: 5 μm. **b** Quantification of endogenous DDB2 accumulation at LUD, normalized to the nuclear background and mock-treated U2OS WT 10 min after UV-C, which was set to 1.0. U2OS cells mock- or VCPi-treated were fixed 10 min, 40 min, 2 h and 8 h after LUD induction. Mean and S.E.M. of, respectively, n = 104,96, 150, 91, 124, 145, 120, 68 cells from two independent experiments. **c** FRAP analysis of GFP-DDB2 mobility in VH10 cells before and immediately after UV irradiation (10 J/m²), in the presence or absence of VCPi (10 μM) added 1 h before irradiation. GFP-DDB2 fluorescence recovery was measured in a strip across the nucleus after bleaching and normalized to the average pre-bleach intensity (1.0). Curves represent the average of >30 cells per condition from three independent experiments. **d**, **e** Recruitment of endogenous (**d**) XPC and (**e**) XPB to LUD in U2OS WT cells in the absence or presence of VCP inhibitor (VCPi), as described in (**a**). Scale bars: 5 μm. **f**, **g** Quantification of endogenous accumulation of (**f**) XPC and (**g**) XPB at LUD as described in (**b**). Mean and S.E.M. of, respectively, n = 206, 291, 290, 348, 234, 226, 146, 72 cells for XPC and n = 145, 93, 140, 119, 144, 161, 139, 48 cells for XPB from three and two independent experiments, respectively. *P < 0.05, **P < 0.01, ***P < 0.001, n.s. non-significant, analyzed by one-way ANOVA (see "Methods"). Source data are provided as a Source Data file.

which showed an increased UV-induced immobilization upon VCPi treatment, suggesting that DDB2 molecules are longer bound to DNA damage (Fig. 4c). Contrary, XPC and XPB accumulation at LUD was delayed and suppressed by VCPi, in particular at early time points (Fig. 4d–g). Interestingly, at these early time points, recruitment of XPC and XPB mirrored that of DDB2, i.e., whenever DDB2 accumulation was higher due to VCPi, XPC and XPB recruitment was lower. It thus appears that prolonged binding of DDB2 to damaged chromatin impairs the early steps of NER, implying that dissociation of DDB2 is required to promote the stable association of XPC and TFIIH with damaged DNA.

Since the VCP segregase has many clients in addition to ubiquitylated DDB2, we tested whether the inhibition of XPC and TFIIH recruitment by VCPi is exclusively dependent on the excessive presence of DDB2 (as part of CRL4^DDB2) at UV-damaged sites. To this end, we generated U2OS DDB2 knockout cells by CRISPR/Cas9-mediated gene disruption and confirmed the absence of DDB2 expression and recruitment to DNA damage by immunoblot and IF (Supplementary Fig. 3a; Fig. 5a, b). Accumulation of both XPC and XPB was impaired in the

absence of DDB2 (Fig. 5c–f), in agreement with the known role of DDB2 in facilitating lesion recognition by XPC[6,12,47]. Importantly, we did not observe any additional effect of VCPi on XPC and XPB accumulation in the DDB2 KO cells (Fig. 5c–f).

To confirm this by FRAP analysis, we generated a GFP-XPB knock-in (KI) MRC-5 human fibroblast cell line, by inserting GFP in front of the endogenous *XPB/ERCC3* gene using CRISPR/Cas9-mediated homology-directed repair (Supplementary Fig. 3b, c). After confirming that the KI cell line behaves as WT MRC-5 in response to UV irradiation (Supplementary Fig. 3d–f), validating the functionality of GFP-tagged XPB, we measured the mobility of this endogenous GFP-XPB in response to UV with and without VCPi, and after depletion of DDB2, using FRAP (Fig. 5g, h). We applied the same approach with recently published GFP-XPC KI HCT116 cell lines that are either DDB2 proficient (WT) or DDB2 KO[48] to measure the impact of VCPi on the mobility of endogenous GFP-XPC in response to UV (Fig. 5i, Supplementary Fig. 3g). UV irradiation led to a strong immobilization of both XPB and XPC, which was partially inhibited by VCPi, corroborating our IF experiments. This inhibition by VCPi was not observed in the absence of DDB2 (Fig. 5g–i, Supplementary

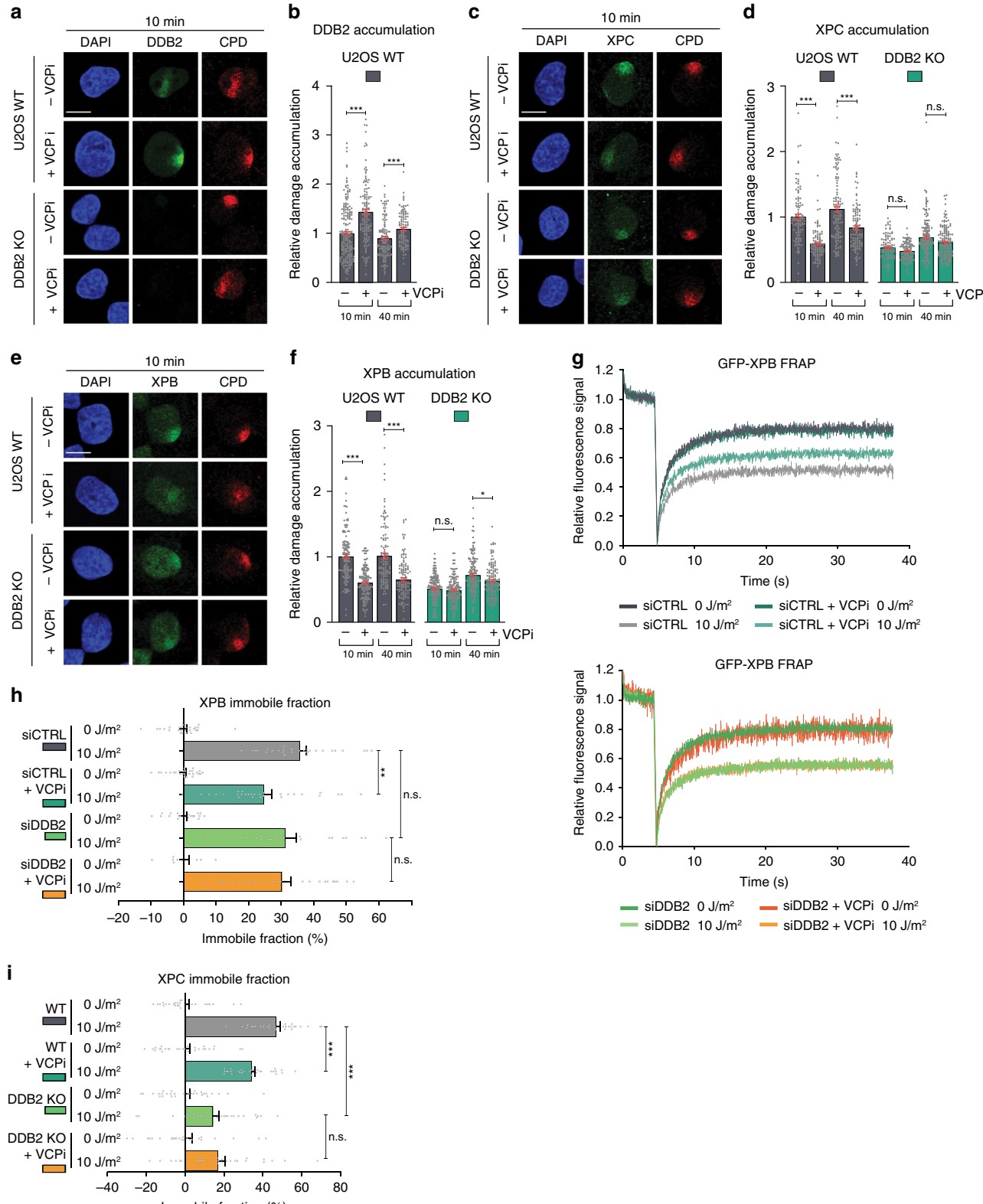

Fig. 3g), unequivocally showing that the reduced XPC and XPB accumulation after VCPi is dependent on DDB2. Previously, it was shown in in vitro cell-free NER excision and reconstituted NER assays that the CRL4DDB2 complex blocks repair in the absence of functional ubiquitylation, because of which it was suggested that ubiquitylation regulates the displacement of DDB2 by XPC at DNA lesions[23]. Together with our data, this supports a

scenario in which the displacement of ubiquitylated DDB2 by VCP promotes damage handover to XPC and the formation of a stably bound damage verification complex together with TFIIH.

**Reciprocal coordination of DNA damage handover in GG-NER.** We expected DDB2 to become more susceptible to

**Fig. 5 VCP-mediated DDB2 extraction facilitates the stable binding of XPC and TFIIH. a** Representative IF images and **b** quantification of endogenous DDB2 accumulation at LUD in U2OS WT and DDB2 KO cells, at the indicated time points after UV irradiation (60 J/m$^2$). Data were normalized to nuclear background and U2OS WT 10 min. Mean and S.E.M. of, respectively, n = 200, 132, 120, 102 cells from two independent experiments. **c** Representative IF and **d** quantification of endogenous XPC accumulation at LUD in U2OS WT and DDB2 KO cells. Mean and S.E.M. of, respectively, $n$ = 106, 93, 117, 103, 97, 101, 148, 150 cells from two independent experiments. **e** Representative IF and **f** quantification of endogenous XPB accumulation at LUD in U2OS WT and DDB2 KO cells. Mean and S.E.M. of, respectively, $n$ = 140, 143, 117, 113, 188, 145, 142, 132 cells from two independent experiments. **g** FRAP analysis of endogenously GFP-tagged XPB mobility before and 1 h after UV irradiation (10 J/m$^2$), in the presence and absence of DDB2 and/or VCP activity. MRC-5 cells with GFP knock-in at the *ERCC3/XPB* locus were transfected with control (CTRL) or DDB2 siRNAs and incubated with mock or VCPi (10 μM). GFP-XPB fluorescence recovery was measured in a strip across the nucleus for 30 s after bleaching and normalized to the average pre-bleach intensity (1.0). **h** Percentage of endogenous XPB immobile fraction in MRC-5 cells treated with CTRL or DDB2 siRNAs and/or VCPi, determined from FRAP analysis as depicted in (**g**). Mean and S.E.M. of >30 cells per condition from three independent experiments. **i** Percentage of endogenous XPC immobile fraction in HCT116 cells with (WT) or without DDB2 (DDB2 KO), mock or VCPi treated, determined from FRAP analysis as depicted in Supplementary Fig. 3g. Mean and S.E.M. of >25 cells per condition from two independent experiments.*$P < 0.05$, **$P < 0.01$, ***$P < 0.001$, n.s. non-significant, analyzed by one-way ANOVA in (**b**), (**d**) and (**f**) and by unpaired, two-tailed $t$-test (adjusted for multiple comparison) in (**h**) and (**i**) (see "Methods"). Scale bars: 5 μm. Source data are provided as a Source Data file.

auto-ubiquitylation by the CRL4$^{DDB2}$ complex due to its increased and continuous recruitment to LUD after treatment with VCPi. However, we found that both VCPi and MG132 treatments strongly suppressed DDB2 ubiquitylation after UV (Supplementary Fig. 4a, b). This explains why in the MG132-treated XPF KO cells, in the absence of repair, DDB2 still accumulated at LUD 8 h after UV irradiation, because DDB2 cannot be proteolytically degraded after UV (Fig. 3f, Supplementary Fig. 2a–c). We also noticed that likely due to depletion of the free ubiquitin pool in cells[49], VCPi prevented efficient UV-induced XPC ubiquitylation (Supplementary Fig. 4c, d), which was hypothesized to increase the affinity of XPC for DNA damage[23,50,51]. Therefore, we devised an alternative strategy to retain DDB2 in damaged chromatin while preserving the functionality of the CRL4$^{DDB2}$ E3 ubiquitin-ligase activity in modifying its substrates, except for DDB2 itself. Previously it has been reported that the N-terminal tail of DDB2 contains several lysines that are targeted for ubiquitylation by the CRL4$^{DDB2}$ complex and are required for degradation of DDB2 after UV-induced damage[21,34]. In addition, structural studies of the CRL4$^{DDB2}$ complex have identified five potential ubiquitylation lysines outside the N-terminal domain (K146, 151, 187, 233, and 278)[21]. Ablation of the first 40 N-terminal amino acids of DDB2 (ΔNT), which include seven lysines (K4, 5, 11, 22, 35, 36, and 40), together with lysine-to-arginine substitutions of the additional five putative ubiquitylated lysines (ΔNT/BP5KR), was shown to inhibit the vast majority of DDB2 UV-induced ubiquitylation in vitro[34]. Therefore, we stably complemented our U2OS DDB2 KO cell line with GFP-tagged full-length WT, ΔNT and ΔNT/BP5KR DDB2 cDNA (Fig. 6a). In contrast to WT GFP-DDB2, both the ΔNT and the ΔNT/BP5KR GFP-DDB2 mutants resisted degradation and were not ubiquitylated after UV irradiation (Fig. 6b, c, Supplementary Fig. 5a). All GFP-DDB2 variants co-immunoprecipitated DDB1, CUL4A, and CSN5 proteins, showing that the assembly of the CRL4$^{DDB2}$ complex is not disturbed by the mutations generated in DDB2 (Fig. 6c). Importantly, the ubiquitylation of XPC after UV irradiation, which is abrogated in DDB2 KO cells, was similarly rescued by WT, ΔNT, and ΔNT/BP5KR GFP-DDB2, indicating that the mutated CRL4$^{DDB2}$ complexes are fully functional (Fig. 6d, Supplementary Fig. 5b, c).

To investigate whether indeed the dissociation of DDB2 from UV-damaged DNA was impeded by the mutations that prevent its ubiquitylation, we measured the residence time of the GFP-DDB2 variants at damaged sites, using inverse fluorescence recovery after photobleaching (iFRAP)[39,52]. To this end, cells were locally irradiated by 266 nm UV-C laser to induce the accumulation of GFP-tagged proteins at the damaged areas. After reaching steady-state accumulation, the nuclear fluorescent signal

was bleached except for the damaged area and a non-damaged control area. Next, the fluorescence decay over time in these two areas was measured (Fig. 6e), which reflects DDB2's residence time in damaged and undamaged chromatin[39]. Accumulation at the laser-induced LUD was higher for the two DDB2 mutants (Supplementary Fig. 5d), and their residence time in damaged chromatin was, on average, 30% increased (Fig. 6f). This shows that ΔNT and ΔNT/BP5KR DDB2 proteins do not efficiently dissociate from DNA lesions, confirming that ubiquitylation facilitates DDB2 displacement. Therefore, using these cell lines, we tested by IF if prolonged DDB2 retention at UV damage inhibits stable XPC binding to DNA damage. In comparison with WT GFP-DDB2 complemented cells, endogenous XPC accumulation at LUD was reduced in the cell lines complemented with either the ΔNT or the ΔNT/BP5KR GFP-DDB2 mutants (Fig. 6g, h, Supplementary Fig. 5e). Moreover, while VCPi treatment did not affect the real-time recruitment of the ΔNT/BP5KR GFP-DDB2 mutant to LUD (Supplementary Fig. 5f), it further inhibited the reduced recruitment of XPC and XPB in cells expressing this mutant (Supplementary Fig. 5g, h). Because VCPi treatment diminishes the levels of UV-induced XPC ubiquitylation (by depletion of the free ubiquitin pool; Supplementary Fig. 4c, d), these observations show that both DDB2 dissociation and ubiquitylation of XPC promote a stable association of XPC and TFIIH to damaged DNA.

Finally, we tested by iFRAP analysis if the recruitment of TFIIH affects the dissociation of DDB2. Interestingly, the depletion of GTF2H1 resulted in prolonged binding of GFP-DDB2 to damaged chromatin (Supplementary Fig. 5i). These data indicate that the increased immobilization of DDB2 after GTF2H1 depletion observed in FRAP (Fig. 1a, b) is caused by prolonged DDB2 binding. Together, our results suggest that the initiation of GG-NER consists of reciprocally coordinated events during which, after the facilitation of UV-damage detection by DDB2, XPC recruits TFIIH, which in turn facilitates the displacement of DDB2 and the stabilization of XPC association with DNA (Fig. 7).

## Discussion

XPC (in complex with CETN2 and RAD23B) is the primary damage sensor of GG-NER and, as such, recruits the TFIIH complex to DNA damage[26,27,53,54] to verify the presence of NER lesions[29,30]. Earlier FRAP studies have suggested that mammalian XPC interrogates DNA integrity through continuous random probing and utilizes a stepwise mechanism to detect and bind DNA damage, in which it first transiently interacts with DNA before forming a stable and immobile damage-bound complex[7,55]. Crystal structures of the yeast XPC ortholog Rad4

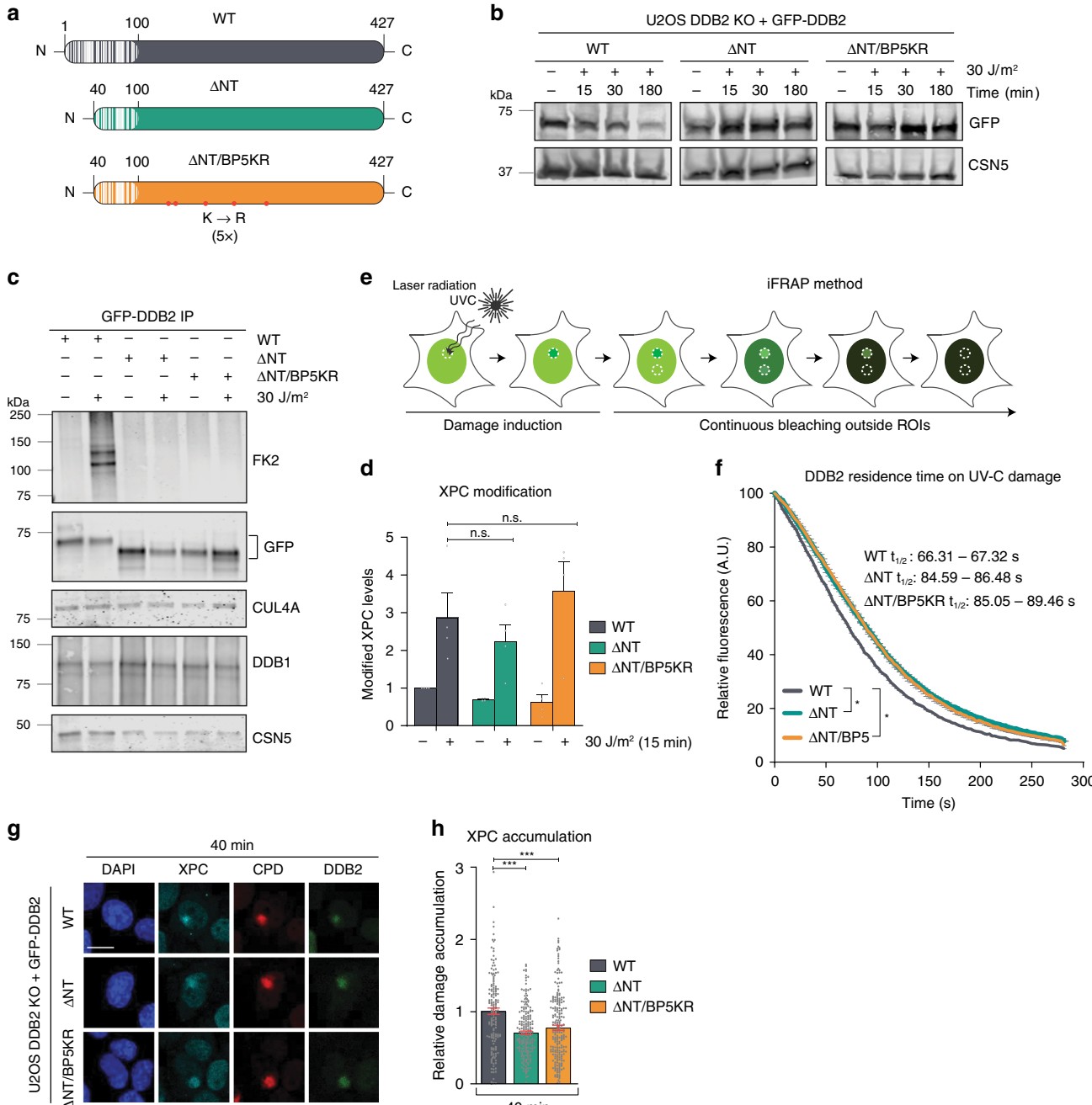

**Fig. 6 DDB2 ubiquitylation facilitates its damage extraction to promote damage handover to XPC. a** Overview of DDB2 wild-type (WT, 427 amino acids) and deletion mutants lacking first 40 amino acids (white stripes; ΔNT) or carrying additionally five lysine to arginine substitutions (red dots; ΔNT/BP5KR). **b** Immunoblot of UV-induced DDB2 proteolysis in U2OS DDB2 KO cells expressing GFP-tagged WT, ΔNT or ΔNT/BP5KR DDB2, analyzed in total cell lysates with DDB2 and CSN5 antibodies. **c** Immunoblot of DDB2 immunoprecipitation showing binding partners and UV-induced ubiquitylation in U2OS DDB2 KO cells expressing WT, ΔNT or ΔNT/BP5KR DDB2, before and 15 min after UV-C irradiation (30 J/m²), analyzed using FK2, GFP, DDB1, CUL4A, and CSN5 antibodies. **d** Quantification of ubiquitylated XPC in whole-cell lysates of U2OS DDB2 KO cells expressing WT, ΔNT or ΔNT/BP5KR GFP-DDB2, analyzed by immunoblot in Supplementary Fig. 4b, c and normalized to Tubulin and mock-treated WT DDB2. Mean and S.E.M. of four independent experiments. **e** Scheme of inverse FRAP (iFRAP) method. Accumulation of a fluorescent protein to local UV-C-laser-induced damage was measured until reaching a steady-state level, after which the GFP-fluorescence outside the UV-damaged and control area was bleached. The loss of fluorescence in the control and UV-damaged areas was measured. **f** iFRAP of WT (gray), ΔNT (green) and ΔNT/BP5KR (orange) GFP-DDB2 dissociation from local UV-damage in U2OS DDB2 KO cells. Fluorescence loss, reflecting DDB2 dissociation, was measured over time, normalized to the background and to fluorescence levels before bleaching. Mean and S.E.M. of >30 cells per condition from three independent experiments. **g** IF images and **h** quantification of endogenous XPC (cyan) accumulation at LUD (CPD, red) in U2OS DDB2 KO cells expressing WT, ΔNT or ΔNT/BP5KR GFP-DDB2 (green), 40 min after UV irradiation (60 J/m²). Data were normalized to the nuclear background and WT. Mean and S.E.M. of, respectively, n = 163, 207, 225 cells from three independent experiments. *P < 0.05, ***P < 0.001, n.s., non-significant, analyzed by unpaired, two-tailed t-test (adjusted for multiple comparisons) in (**d**), by ROC curve analysis in (**f**) and by one-way ANOVA in (**h**) (see "Methods"). Scale bars: 5 μm. Source data are provided as a Source Data file.

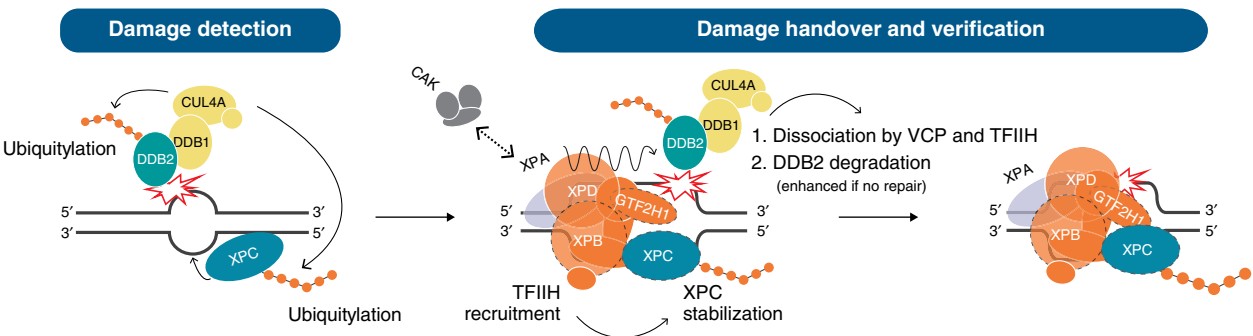

**Fig. 7 Reciprocal coordination of DNA damage detection and handover in GG-NER.** DDB2 binds directly to UV-photoproducts, thereby stimulating XPC recruitment to CPDs and 6-4PPs. The CRL4 E3 ubiquitin ligase is activated upon DDB2 binding and ubiquitylates DDB2 and XPC. TFIIH is recruited via an interaction between its subunit XPB with XPC (interaction depicted with dotted lines). Upon TFIIH binding, its trimeric CDK7-activating kinase (CAK) sub-complex is released and allows XPA binding, which further stimulates TFIIH's XPD helicase that unwinds the DNA in the 5′–3′ direction while scanning for helicase blocking lesions. This configuration facilitates further interaction between TFIIH and XPC by allowing GTF2H1 to interact with XPC. Recruitment of TFIIH and ensuing damage verification promote the stable association of XPC with the undamaged strand and simultaneously facilitate the displacement of DDB2, which is also promoted by ubiquitylation-mediated extraction by VCP (1). The subsequent degradation of DDB2 (2) regulates its availability to rebind to lesions, possibly to avoid competition with the emerging NER pre-incision complex. The formation of a stable ternary XPC-TFIIH-XPA damage verification complex on the lesion and the unpaired DNA surrounding the lesion (created by this complex) provide substrate for the structure-specific endonucleases XPF-ERRC1 and XPG (the latter coinciding with XPC dissociation), which completes the formation of the pre-incision complex.

bound to non-damaged DNA[8], CPD or 6-4PP photolesions[11,56], recent in vitro temperature-jump spectroscopy[9] and single-molecule imaging on yeast and human XPC[57–59], together with computational modeling, point to a model in which damage recognition by XPC is characterized by consecutive stages: (1) a search complex with random motion; (2) a transiently stalled interrogation complex that untwists and bends the DNA, due to the insertion of XPC's BHD2 hairpin in the minor groove that opens the DNA around the lesion; and (3) a final recognition complex fully and stably bound to the DNA due to the insertion of XPC's BHD3 hairpin into the major groove at the lesion site without ever contacting the lesion directly[6,10]. In this model, the capacity of XPC to recognize a lesion is dependent on its ability to open the damaged dsDNA and insert its BHD3 hairpin before diffusing away[8]. Strikingly, we found that in living cells, TFIIH is required for stable binding of XPC to damaged DNA (Figs. 1c, d, 3c). This suggests that TFIIH recruitment may either stabilize the transient interrogation complex, thus promoting the transition to a fully immobile recognition complex, or stabilize the recognition complex itself, by preventing reversion back to an interrogation complex. Accordingly, previous in vitro DNA-binding studies have suggested that upon DNA binding, XPC can form a stable ternary complex with TFIIH and XPA that is even able to translocate along DNA[29]. Also, recent modeling analysis based on the structural resolution of XPC and TFIIH indicates that damage verification by TFIIH can stabilize its interaction with XPC on DNA[10]. This model proposes that TFIIH is recruited to DNA through an interaction between its XPB subunit and the XPC C-terminus. Upon the release of the CAK sub-complex from TFIIH, stimulated by XPA, the TFIIH helicase XPD contacts the DNA and translocates on the damaged DNA strand in a 5′ to 3′ direction until it is blocked by a lesion, i.e., damage verification. In this conformation, the TFIIH subunit GTF2H1 is then able to interact with the N-terminus of XPC. Interestingly, XPA enhances lesion-scanning by TFIIH[30,60] and we found by FRAP that, like TFIIH, XPA facilitates stable binding of XPC to UV-damaged DNA (Supplementary Fig. 6). Therefore, we propose that the formation of a stable XPC-TFIIH-DNA complex is stimulated by active damage verification activity and not solely by the recruitment of TFIIH.

The energetic barrier for XPC to open the dsDNA and form a stable XPC-DNA recognition complex is higher for lesions that do not strongly distort the DNA duplex[61]. This explains the much lower affinity of XPC for CPDs, which only minimally distort the DNA, as compared to the more helix-destabilizing 6-4PPs[17]. DDB2 assists XPC in recognizing UV-induced lesions by directly binding the lesions[15,16] and is thus more relevant for CPDs, albeit it stimulates the repair of 6-4PPs in vivo as well[34,62]. Contrary to XPC, in the absence of TFIIH, we observed increased binding and recruitment of DDB2 to local UV-induced damage (Fig. 3a, b). Moreover, using iFRAP, we measured prolonged DDB2 retention at lesion sites (Supplementary Fig. 5i), suggesting that DDB2 dissociation coincides with TFIIH recruitment and the stabilization of the XPC-TFIIH-DNA complex. Previously, it was shown that tethering DDB2 to chromatin recruits XPC but never TFIIH, whereas tethering XPC recruits TFIIH but never DDB2, implying that DDB2 and TFIIH associate with XPC on DNA damage in a mutually exclusive manner[63]. Furthermore, the superimposition of the crystal structures of DDB2 and yeast XPC/Rad4 bound to DNA indicates that the two proteins cannot stably bind the same lesion simultaneously, as both interact with the DNA minor groove around the lesion[10,11,15,56]. However, lesion-bound CRL4[DDB2] is required for XPC ubiquitylation[23], arguing that DDB2 and XPC should—temporarily—coexist, prior to the handover of the damage to the XPC-TFIIH verification complex. Furthermore, XPC uses separate domains to bind to DNA adjacent to and opposite of the lesion in a stepwise manner[11,55,56]. We thus envision that when XPC is recruited to DNA damage, DDB2 and TFIIH exchange to promote its stable binding. In this scenario, TFIIH recruitment to XPC and binding to DNA stimulates DDB2 release and, hence, the transition of XPC from an interrogation to a stably bound recognition complex.

In compliance with this hypothesis, it was found that in vitro reconstituted NER of 6-4PPs is inhibited by the addition of excess DDB2 in the absence of ubiquitylation factors that mediate its release[14,23]. Moreover, here we observed that also in living cells when DDB2 is retained at DNA lesions, recruitment of XPC and XPB is inhibited (Figs. 5, 6). Altogether, these results imply that excessive DDB2, e.g., its prolonged binding, can impede the stable binding of subsequent NER factors. Interestingly, structural studies have indicated that the UV-DDB complex can form tightly DNA-bound dimers, which appears to be concentration dependent[16,64] and could, therefore, also be involved in the inhibition of repair by excess DDB2.

We found that unrepaired lesions, i.e., after the loss of XPG or XPF, lead to persistent DNA damage sensing by DDB2 and XPC (Figs. 1, 2), similar to the persistent targeting of the core NER machinery to DNA damage after the loss of functional XPF that we described before[45]. XPC is believed to dissociate from DNA lesions concomitantly with XPG recruitment[39,65]. Therefore, its increased and persistent accumulation in XPF KO cells (Fig. 2d, e) likely reflects continuous binding to and dissociation from lesions that remain accessible. In the case of DDB2, this continuous binding to and dissociation from DNA lesions causes an accelerated UV-induced degradation, rescued by proteasome inhibition (Fig. 2a–c; Supplementary Fig. 2a–c). It was previously estimated, based on photobleaching experiments, that DDB2 can rebind DNA damage multiple times before being degraded[43]. Combined with the fact that most other NER proteins, like XPC, are not degraded after UV, this indicates that the effective DDB2 concentration must be tightly regulated in order to promote proper handover of damage to XPC and TFIIH.

Ubiquitylation plays a key role in controlling DDB2 association with lesions, both by lowering its affinity towards DNA[23,34] as well as by lowering its protein concentration through degradation[20,22]. Besides, ubiquitylated DDB2 is actively extracted from chromatin by the VCP segregase, which was shown to facilitate DNA repair and to prevent chromosomal aberrations[34,35]. Here, we show that impairing DDB2 dissociation, by inhibiting VCP activity or mutating the DDB2 ubiquitylated lysine residues, compromises recruitment of the downstream NER machinery to lesions. Nonetheless, we still observed DDB2 dissociation from damage in VCP-inhibited cells, albeit delayed (Fig. 4a, b). A similar delayed release from damaged chromatin was previously observed with DDB2 lysine mutants, implying that ubiquitylation promotes but is not essential for DDB2 dissociation[34]. Additionally, we found that inhibition of UV-induced DDB2 degradation by MG132 treatment did not prevent its release from damage in NER-proficient cells and allowed DDB2 to rebind persistent lesions over time in NER deficient cells (Supplementary Fig. 2a, b). Hence, the degradation of DDB2 regulates its availability to recognize and bind to damaged DNA and is separate from its extraction and dissociation from DNA. As DDB2 has a stronger affinity for UV photolesions than XPC[13,66], its degradation likely prevents that too many DDB2 molecules are available to rebind the same lesions. These results suggest that similar to the recruitment of TFIIH, DDB2 ubiquitylation promotes proper DNA damage handover and the formation of a stable XPC-TFIIH-DNA lesion verification complex.

In summary, here we present evidence of a dynamic interplay between NER DNA damage sensors DDB2 and XPC and the TFIIH verification complex. Based on our findings and relevant literature, we propose that the following key events take place in the transition from damage detection to verification (see also Fig. 7). First, DDB2 binds directly to UV-photolesions and stimulates the recruitment of XPC. Ubiquitylation (by CRL4$^{DDB2}$) of DDB2 reduces its affinity towards UV-lesions and accelerates its dissociation via extraction by VCP. Dissociated ubiquitylated DDB2 is targeted for proteasomal degradation, which decreases its effective concentration. When more DDB2 molecules bind to lesions, e.g. in case of deficient NER or higher DNA damage load, more molecules are ubiquitylated and thus proteasomal degradation is enhanced. Upon XPC recruitment, also TFIIH is recruited via interaction with XPB, which coincides with or even stimulates the dissociation of DDB2. Possibly, DDB2 displacement is facilitated by physical competition for the binding space in the vicinity of the lesion or by TFIIH's translocation activity. Ubiquitylation of XPC (by CRL4$^{DDB2}$) increases its affinity for DNA damage while TFIIH recruitment, likely due to the XPA-

stimulated activation of its helicase activity, stabilizes XPC DNA binding through the formation of an XPC-TFIIH-DNA complex via an additional interaction between XPC and GTF2H1 (Fig. 7). Besides ubiquitylation, many more PTMs have been reported to control DDB2 and XPC activity, including PARylation, sumoylation and phosphorylation[36,38–42,67,68]. Therefore, it would be interesting to investigate in the future how these PTMs may be controlling the dynamic damage handover between NER initiation and verification factors.

## Methods

**Cells lines, culture conditions, and treatments**. U2OS WT, DDB2 KO and XPF KO[45], SV40-immortalized human fibroblasts XP4PA (XPC-deficient, with stable expression of XPC-GFP), hTERT-immortalized human fibroblasts VH10 (with stable expression of GFP-DDB2[41] or GFP), HCT116 (with GFP-XPC KI)[48] and MRC-5 (with GFP-XPB KI) were cultured at 37 °C in a humidified atmosphere with 5% CO$_2$ in a 1:1 mixture of DMEM (Lonza) and Ham's F10 (Lonza) supplemented with 10% fetal calf serum (FCS) and 1% penicillin-streptomycin. XP4PA cells with stable expression of XPC-GFP were generated using lentiviral transduction and selection with 0.3 μg/mL Puromycin and FACS[69]. To generate GFP-XPB KI cells, MRC-5 cells were transiently transfected with pLentiCRISPR-v2[70] carrying a sgRNA targeting near the START codon of the XPB/ERCC3 locus, and pCRBluntIITOPO carrying GFP cDNA flanked by XPB homology sequences. After selection with 2 μg/mL Puromycin and FACS, a clonal cell line was isolated and verified by sequencing and functional analysis (Supplementary Fig. 3b–f). To generate U2OS DDB2 KO cells, U2OS cells were transiently transfected with pLentiCRISPR-v2[70] containing a sgRNA targeting near the START codon of the DDB2 locus. Transfected cells were selected with puromycin and a correct DDB2 KO clone was isolated and verified by sequencing and functional analysis (Fig. 5a, Supplementary Fig. 3a). U2OS DDB2 KO cells with stable expression of WT, ΔNT or ΔNT/BP5KR GFP-DDB2 cDNA were generated using lentiviral transduction and selection with 10 μg/mL Blasticidin and FACS. siRNA transfections were carried out 48 h before each experiment using RNAiMax (Invitrogen) according to the manufacturer's instructions. Plasmid transfections were performed using JetPei (Promega), according to the manufacturer's instructions. To inhibit proteasome or VCP activity, cells were treated with 50 μM MG132 (BML-PI102, Enzo) or 10 μM of VCPi (NMS-873, Selleckchem), respectively, 1 h before UV irradiation.

**Plasmids, sgRNA, and siRNA**. To generate an XPC-GFP plasmid, full-length human XPC cDNA was fused to GFP and inserted into pLenti-CMV-Puro-DEST[69]. The pLenti6.3 WT GFP-DDB2 plasmid was kindly provided by Dr. A. Pines[41]. ΔNT and ΔNT/BP5KR GFP-DDB2 plasmids were generated by deleting the first N-terminal 120 base pairs of DDB2 (ΔNT) and inserting a DDB2 fragment containing five lysine to arginine substitutions (BP5KR) from plasmid pIREShyg-HA-DDB2-NdeI/BP5KR[34], which was a kind gift from Dr. K. Sugasawa. The sgRNAs targeting the XPB/ERCC3 (TCTGCTGCTGTAGCTGCCAT) and DDB2 (CACCGCCTTCACACGGAGGACGCGA) loci were cloned into pLenti-CRISPR-V2[70]. The homologous repair template, with GFP DNA flanked by XPB sequences, was generated by PCR (using primers Frw1_HA_XPB_Nt: GCGGATGCCGCGCG CGGGCCTGTGGGAGCGGGGTCATCTTCTCTCTGCTGCTGTAGCTGCCAT GATTGTGAGCAAGGGCGAGGAGCT and Rv1_HA_XPB_Nt: CAGTCGTGG CTGAGCGTGCCCGCGCAACGTCTCACCGCGGTCCGCTCGGTCTCTTTT GCCCTTGTACAGCTCGTCCATGC) and cloned into the pCRBluntIITOPO vector (Zero Blunt$^{TM}$ TOPO$^{TM}$ PCR Cloning Kit, ThermoFischer Scientific). Additional cloning and plasmid details are available upon request. siRNA oligomers were purchased from GE Healthcare: CTRL (D-001210-05), DDB2 (J-011022-05), XPG (M-006626-01) and GTF2H1 (L-010924-00). siRNA knockdown efficiency was tested by western blot or IF for each experiment, as shown in Supplementary Fig. 1.

**UV-C irradiation**. Using a germicidal lamp (254 nm; TUV lamp, Phillips), cells were UV-C irradiated with the indicated doses after being washed with PBS. Local UV-damage (LUD) was generated by applying 60 J/m$^2$ of UV irradiation through an 8 μm polycarbonate filter (Millipore) that was placed on top of a monolayer of cells[69].

**Immunofluorescence**. Cells were grown on 18 mm coverslips, fixed in 4% paraformaldehyde, and permeabilized in PBS containing 0.5% Triton X-100. For visualization of local UV-induced DNA damage (LUD), DNA was denatured for 5 min with 70 mM NaOH. Next, cells were incubated in blocking buffer (3% BSA and 2.25% glycine in PBS-T (0.1% Tween 20)) for 1 h at room temperature. Primary antibodies were incubated for 1–2 h at room temperature or overnight at 4 °C and secondary antibodies conjugated to Alexa fluorochromes 488 or 555 (Invitrogen) were incubated for 1 h at room temperature. The antibody incubation solution was 1% BSA in PBS-T. DNA was stained with DAPI (Sigma), and slides were mounted using Aqua-Poly/Mount (Polysciences, Inc.). Antibodies used are summarized in Supplementary Tables 1 and 2. Image acquisition was performed

using an LSM700 microscope equipped with a 40x Plan-apochromat 1.3 NA oil immersion lens (Carl Zeiss Micro Imaging Inc.). To quantify protein recruitment to lesion sites, the fluorescence signal intensity at LUD was divided by the nuclear intensity, as measured using FIJI image analysis software (version 1.52p). Zero accumulation (nuclear background) was set to 0 and maximum accumulation (above nuclear background) in control or mock-treated conditions was set at 1.0.

**Immunoprecipitation (IP)**. IP experiments were performed under denaturing conditions to detect DDB2 modifications. VH10 GFP-DDB2 cells were grown to confluency on 10 cm dishes and lysed 15 min after UV-C irradiation (30 J/m$^2$) in lysis buffer (20 mM Tris-HCl pH 7.5, 50 mM NaCl, 0.5% NP-40, 1% SDS, 5 mM MgCl$_2$ and EDTA-free protease inhibitor cocktail (Roche)). Cell lysates were incubated with benzonase buffer (20 mM Tris-HCl pH 7.5, 50 mM NaCl, 0.5% NP-40, 0.5% Sodium Deoxycholate, 0.5% SDS, EDTA-free protease inhibitor cocktail (Roche) and 0.25 U/μL Benzonase (Millipore)) for 45 min at room temperature in a tube rotator for digestion of chromatin. The suspension was spun down (15.000 g for 10 min) and the supernatant (Input) was used for GFP-DDB2 IP (GFP-DDB2 IP), by incubation of GFP-trap beads (Chromotek) for 2 h at room temperature. Beads were washed 5× (20 mM Tris-HCl pH 7.5, 50 mM NaCl, 0.5% NP-40, 0.5% Sodium Deoxycholate, 0.5% SDS and EDTA-free protease inhibitor cocktail (Roche)) and elution of immunoprecipitated proteins was performed by boiling the GFP-trap beads in 2× sample buffer for 5 min at 98 °C. Input and GFP-DDB2 IP fractions were analyzed by immunoblotting.

**Fluorescence recovery after photobleaching (FRAP)**. For FRAP analysis analysis[69,71], the GFP-fluorescence signal of our GFP-tagged proteins was measured in a strip across the nucleus (width 512 × 16 pixels, zoom ×12), at 1400 Hz of a 488 nm laser every 22 ms until a steady-state was reached (pre-bleach). Using 100% power of the 488 nm laser, the fluorescent signal in the strip was bleached and fluorescence recovery was monitored every 22 ms until recovery was complete. All FRAP experiments were acquired on a Leica TCS SP5 microscope (with LAS AF software, Leica, version 2.7.4.10100) equipped with a 40x/1.25 NA HCX PL APO CS oil immersion lens (Leica Microsystems), at 37 °C and 5% CO$_2$. Fluorescence signals were normalized to the average pre-bleach fluorescence after background signal subtraction. For the quantification of the immobile fractions ($F_{imm}$), shown in Fig. 1b, d; 5h, i; Supplementary Fig. 1j, 6, the average recovered fluorescence intensity of UV-irradiated cells ($I_{final,UV}$) was divided by the average recovered fluorescence intensity of unchallenged cells ($I_{final,unc}$) over the last 10 s of the measurements, after correction with the fluorescence intensity recorded immediately after bleaching ($I_0$)[69]:

$$Fimm = 1 - \frac{Ifinal, UV - I0, UV}{Ifinal, unc - I0, UV}. \tag{1}$$

**UV-C laser accumulation and inverse FRAP**. Accumulation of proteins to UV-C laser-induced DNA damage was measured on a Leica SP5 confocal microscope (with LAS AF software, Leica, version 2.7.4.10100) coupled to a 2 mW pulsed (7.8 kHz) diode-pumped solid-state laser emitting at 266 nm (Rapp Opto Electronic, Hamburg GmbH; Supplementary Fig. 5d) or on a Leica SP8 confocal microscope (with LAS X software version 3.5.6.21594), coupled to a 4.5 mW pulsed (15 kHz) diode-pumped solid-state laser emitting at 266 nm (Rapp Opto Electronic, Hamburg GmbH; Supplementary Fig. 5f). Cells, grown on quartz coverslips, were imaged and irradiated through an Ultrafluar quartz 100×/1.35 NA glycerol immersion lens (Carl Zeiss Micro Imaging Inc.) at 37 °C and 5% CO$_2$. Resulting accumulation curves were corrected for background values and normalized to the relative fluorescence signal before local irradiation. iFRAP[39,52] was performed on a Leica SP5 confocal microscope by bleaching the entire nucleus after accumulation reached a steady-state level accumulation, except for three areas in which the fluorescence decay was measured over time: the area of laser-induced UV-C damage, a non-damaged nuclear area, and a cytoplasmic area (background). After background correction, signals in the damaged and non-damaged areas of the nucleus were normalized to the average fluorescence levels of pre-damage conditions. The half-time of protein residence in the damaged area was determined by applying a non-linear regression fitted to one-phase exponential decay analysis to the iFRAP curves (Fig. 6f), using Graph Pad Prism version 8.21 for Windows (GraphPad Software, La Jolla California USA).

**Preparation of total cell extracts**. Cells were washed twice in ice-cold PBS and lysed on ice for 15 min in RIPA buffer (25 mM Tris-HCl pH 8.0, 150 mM NaCl, 0.1% SDS, 1% NP-40, 0.5% Sodium Deoxycholate, 5 mM EDTA, 1 mM PMSF and EDTA-free protease inhibitor cocktail (Roche)). Soluble extracts were obtained by centrifugation at 14,000 × g for 30 min at 4 °C and equal protein amounts were diluted in 2× sample buffer for immunoblot analysis. 20 mM of N-ethylmaleimide (E3876, Sigma) (DUB inhibitor) was added to the RIPA buffer to improve visualization of XPC-ubiquitination bands (after UV)[72].

**Immunoblotting**. Protein samples (total cell extracts or IP fractions) were 2× diluted in sample buffer (125 mM Tris-HCl pH 6.8, 20% Glycerol, 10% 2-β-Mercaptoethanol, 4% SDS, 0.01% Bromophenol Blue) and boiled for 5 min at 98 °C.

Proteins were separated in SDS-PAGE gels and transferred onto PVDF membranes (0.45 μm, Merck Millipore). One hour after blocking the membranes in 5% BSA in PBS-T (0.05% Tween 20), primary antibodies (in PBS-T) were added for 1–2 h at room temperature, or 4 °C overnight. Secondary antibodies were incubated for 1 h at room temperature. After each step of antibody incubation, membranes were washed 3 × 10 min in PBS-T. Probed membranes were visualized and densitometrically analyzed with the Odyssey CLx Infrared Imaging System (LI-COR Biosciences). Antibodies are listed in Supplementary Tables 1 and 2.

**Statistical analysis**. Mean values and S.E.M. error bars are shown for each experiment. Multiple t-tests (unpaired, two-tailed) were used to determine statistical significance between groups followed by multiple comparison correction with the Holm-Sidak method without assuming a consistent standard deviation. For the statistical significance analysis of IF data, we applied a One-Way ANOVA using the Brown-Forsythe and Welch ANOVA tests, followed by post-hoc analysis with the Games-Howell method. For analysis of graphs in Fig. 6f and Supplementary Fig. 5f, i a ROC curve analysis was performed with significance levels set to 0.05. All analyses were performed using Graph Pad Prism version 8.21 for Windows (GraphPad Software, La Jolla California USA). P values expressed as *$P < 0.05$; **$P < 0.01$, ***$P < 0.001$ were considered to be significant. n.s, non-significant.

**Reporting summary**. Further information on research design is available in the Nature Research Reporting Summary linked to this article.

## Data availability
Source data underlying Figs. 1–6 and all Supplementary Figs. are provided as a Source Data file with this paper. Any other data are available from the corresponding author upon reasonable request. Source data are provided with this paper.

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

## Acknowledgements

The authors would like to thank Dr. A. Pines for technical help and kindly providing the pLenti6.3 GFP-DDB2 plasmid and Dr. K. Sugasawa for the pIREShyg-HA-DDB2-Ndel/BP5KR plasmid and the Erasmus MC Optical Imaging Center for microscopy support.

This work was financially supported by a Marie Curie Initial Training Network funded by the European Commission 7th Framework Programme (grant 316390), a European Research Council Advanced Grant (grant 340988-ERC-ID), a Worldwide Cancer Research Award (grant 15–1274), Dutch Scientific Organization (ALW grant 854.11.002 and VICI grant VI.C.182.025) and the Dutch Cancer Society (KWF grant 10506) and by the gravitation program CancerGenomiCs.nl from the Netherlands Organization for Scientific Research. This work is part of the Oncode Institute which is partly financed by the Dutch Cancer Society.

## Author contributions

C.R.S., H.L., A.H., and A.F.T. performed all experiments. M.S. and J.A.M. contributed reagents. C.R.S., H.L., and W.V. designed experiments, analyzed data and wrote the manuscript. All authors reviewed the manuscript.

## Competing interests

The authors declare no competing interests.
