## [Peer Review File · Nature Communications]

Reviewers' Comments:

Reviewer #1:

Remarks to the Author:

The dynamic hand-off of DNA damage from one recognition complex to another damage verification protein or set of proteins is an important area of research in the nucleotide excision repair pathway. In this outstanding study the authors explore the hand-off of UV damage from UV-DDB to XPC and TFIIH and the role that ubiquitylation and DDB2 degradation plays in this process. The authors have previously studied the kinetics of UV-DDB and XPC recruitment in several previous papers, but this present study uses state-of-the-art knockout and knock-in approaches provided by CRISPR/Cas9 technologies to extend this work in a highly significant manner as they can follow tagged proteins that are expressed at endogenous levels. They first show that inhibition of later stages of NER by suppressing the incision steps through the removal of one of the endonucleases, leads to longer lived DDB2 and XPC at damaged sites. However, this long lived DDB2 leads to increased DDB2 ubiquitylation. Knockdown of p62 of TFIIH further increased accumulation of DDB2 at damaged in XPC KO cells, but also lead to increased ubiquitylation and degradation of DDB2. Importantly, the authors show that this degradation of DDB2 was not observed in WT cells treated with locally induced UV damage. They next show that DDB2 KO causes less stabilization of XPC and increase recruitment of TFIIH, suggesting that ubiquitylation of XPC by UV-DDB-CUL4A-RBX is important for stabilization. These data taken together strongly suggest a working model that UV-DDB is rapidly recruited to damage sites and helps recruit XPC, subsequent UV-DDB dissociation is facilitated by TFIIH arrival, which allows UV-DDB to move on to recruit XPC to other sites. Under these conditions DDB2, due its relatively short-lived time at damage sites is not highly ubiquitylated and not degraded. Furthermore, these data suggest that only when NER is unbalanced either by slowing later incision events or by overwhelming the repair system with high levels of global damage is DDB2 auto-ubiquitylated and degraded. They further support this their working model with an inhibitor of VCP/p97, which normally helps extract ubiquitylated DDB2 from chromatin. Addition of the inhibitor caused delayed XPC and XPB accumulation at locally damage sites. KO of DDB2 also caused slower XPC and XPC accumulation and the important control of the VCPi had no further effect. The final and most important experiment in this study the authors expressed a N-terminally 40 amino acid truncated DDB2 and mutated an additional five Lys to Arg to block ubiquitylation in a DDB2 KO cell line. This construct and the N-terminally truncated variant was capable of ubiquitylating XPC, but because it itself could not be auto-ubiquitylated remained on DNA longer and thus decreased the recruitment of XPC. In total these exciting data suggest an intricate cellular timer mechanism that under normal conditions DDB2 helps recruit XPC and ubiquitylate it stabilizing at damaged sites, but if DDB2 persists at damaged sites too long, it is turned over by auto-ubiquitylation and degradation. Overall this is a timely and important contribution to the field. The experiments are well designed and executed. This already strong study would benefit from the authors consideration of the following points, which are mostly minor.

1. In the introduction, the authors should cite the important work by Lan et al. (J Biol Chem, 287 (2012) 12036-12049) showed that UV-DDB-CUL4A-RBX binds to nucleosomes containing photoproducts, ubiquitylating H2A at Lys 119 which causes destabilization of the nucleosome.
2. The color scheme in the FRAP traces in Figure 1a & c are difficult to follow; please use different colors.
3. Figure 1d, what happens in a double knockdown experiment, does XPC immobilization stay low without TFIIH recruitment even in the absence of XPG? Would DDB2 remain bound?
4. It would be good to show control westerns indicating the level of knockdown achieved for each siRNA used. For example, GTF2H1 siRNA KD is shown in Figure 1, but the western is not shown until Figure 3. It does not look like XPG or DDB2 levels were ever shown after KD with siRNA. Also are there accompanying westerns for DDB2 and a control protein for the experiments described in 4?
5. Lines 296-297 the authors state that XPC searches with "random motion". In fact, two single molecule studies cited by the authors show anomalous diffusion of Rad4 and XPC around CPD. This anomalous diffusion could provide dynamic recognition of helping to recruit both UV-DDB and

TFIIH to the vicinity of the lesion site without binding directly to the damaged site, which might interfere with subsequent repair events. Thus, at the scale of single molecules binding events might be in fact transient and dynamic and the FRAP experiments described here which are bulk analysis and not able to provide such spatial and temporal resolution. The authors actually revisit this issue again on lines 321-323.

6. To this end, perhaps proximity ligation assays could be used to show direct interaction of DDB2-XPC or XPC-TFIIH, but not DDB2-TFIIH.

7. The discussion provides an exciting model that is consistent with their data. However, it suffers in places, by preferential citation of some, but not all of the literature – this needs correcting. For example:

- Lines 324-325, The model that UV-DDB kinks the DNA duplex while supported by the Thoma structure (ref 15) is not supported by the Yeh structure (Proc Natl Acad Sci U S A, 109 (2012) E2737-2746) which showed little or no kink.

- Lines 334-336, Simultaneous binding of XPC and UV-DDB was modeled by Thoma's group in their Scrima Cell paper (ref 15), see supplemental figure S7 where a co-complex is modeled.

- Line 344-345, That NER is inhibited by addition of DDB2 is inconsistent with work by Matsunaga and coworkers, who using a fully reconstituted NER system consisting of seven purified proteins and 136 base pair (bp) DNA substrates containing either a CPD or 6-4PP lesion, demonstrated that UV-DDB could stimulate excision of CPD by 5-17-fold in a reaction containing 571 fmol of UV-DDB, but only displayed a ~2 fold stimulation for 6-4PP even when used at a significantly reduced amount (0.57 – 5.7 fmol) J Biol Chem, 277 (2002) 1637-1640.). It is also interesting to note that UV-DDB was found by in the crystal structure and by EM and AFM to form dimers of dimers in the Yeh et al paper cited above. Perhaps these dimers of dimers of UV-DDB, which bind very tightly to the DNA are inhibitor to repair and help explain the lack of congruence between the JBC 2002 paper by the Matsunaga group and the Cell 121, 387–400 (2005) by Sugawara et al.

8. The working model shown in Figure 7 is helpful, but not entirely consistent with their own data. For example, the authors state on lines 388-389, "Dissociated ubiquitylated DDB2 is targeted for proteasomal degradation, which decreases its effective concentration." This is only true when XPF is knocked out or XPG is knocked down and as stated on lines 182-184, "Our observations suggest that the recruitment of

TFIIH promotes the stable binding of XPC to damaged DNA and the dissociation of DDB2, thereby preventing excessive DDB2 auto-ubiquitylation and degradation." Thus, the authors should include a caveat to their figure that only during unbalanced NER (i.e. decrease TFIIH recruitment or loss of incision or high damaged levels) does UV-DDB become highly ubiquitylated to such a degree that it is degraded in an appreciable manner. The UV-DDB direct ubiquitylation of XPC and its auto-ubiquitylation is three-fold, helping to recruit XPC at damaged sites and allow UV-DDB dissociation and subsequent TFIIH recruitment. This study and their working model thus helps resolve a long-standing question in the field as to why a cell might kill its first responder, UV-DDB, through protein degradation. Furthermore, one might imagine a DUB that helps deubiquitylate DDB2 after its dissociation and allowing DDB2 to undergo another round of damage recognition and XPC loading.

9. While this might be beyond the scope of the present work, where is ubiquitylated DDB2 being degraded? Within the nucleus or in the cytoplasm? Is it known if the nucleus has active 26S proteasomes. Some speculation in the discussion about this matter would be helpful to the reader.

Reviewer #2:

Remarks to the Author:

In this manuscript, the authors focus on mechanistic regulation of the DNA damage recognition process in mammalian global genomic nucleotide excision repair (GG-NER). For efficient GG-NER of UV-induced DNA damage, coordinated actions of multiple protein factors, including DDB1-DDB2 (UV-DDB), XPC, TFIIH, etc., are crucial. Regulation of this process involves protein-protein and protein-DNA interactions as well as post-translational modifications such as ubiquitination, which

has not yet been fully understood.

Using the well-established technique of fluorescence recovery after photobleaching (FRAP), the authors first show that UV-induced immobilization of DDB2 (likely reflecting association with DNA lesion sites) is enhanced by depletion of a TFIIH subunit (GTF2H1/p62), while immobilization of XPC appears to be compromised. Based on these findings, they present evidence that recruitment of TFIIH facilitates dissociation of UV-DDB from DNA damage sites and rather stabilizes the XPC lesion recognition complex. If proper recruitment of TFIIH or the following DNA incision is compromised, DDB2 is markedly destabilized presumably through enhanced auto-ubiquitination by the associating CUL4A E3 ligase. With additional experiments using some inhibitors and ubiquitination-resistant DDB2 mutants, they conclude that timely ubiquitination of DDB2 and its removal from DNA damage sites are important for efficient damage handover to XPC and the following GG-NER process.

Overall, the manuscript is well written, providing data from experiments that are appropriately designed and performed with high technical quality. The conclusions they made are mostly convincing and supported by the presented results. This reviewer has only minor comments and suggestions, which the authors could consider for further improvement.

1. Some FRAP data are difficult to see, because of overlapping lines (especially Fig. 5g). The authors could consider to change colors of lines and/or separate a graph into multiple ones.
2. Fig. 1b: Title of the graph should be corrected (DDB2).
3. Fig. 3d: This data would be even more convincing, if the authors could include results with control cells lacking GFP-DDB2 expression and show that the UV-induced FK2 signals are not detected.
4. Fig. 6c: This reviewer expects that, after UV irradiation, part of CUL4A undergoes conjugation to NEDD8 and shows a band shift. In addition, amounts of DDB1 in IP fractions do not seem to correlate with GFP signals (especially, in the right most lanes). Does this suggest instability of the complex?
5. Suppl. fig. 2a: This Western blot shows doublet bands of DDB2, which seems different from other DDB2 blots. Could the authors assign each of these species?
6. Suppl. fig. 4: According to the figure legend, panel e seems missing, and the graph labeled "e" should be panel f.

Reviewer #3:

Remarks to the Author:

This manuscript by Ribeiro-Silva describes cellular fluorescence imaging studies that shed light on the mechanism of progression in NER during damage recognition and verification. While the step-wise biochemical processes of global genome NER have been known for a while, our understanding of how the seemingly sequential steps are coordinated in cells has been relatively poor. There are not only multiple protein factors working within chromatinized DNA environment but also multiple distinct posttranslational modifications for the proteins factors that play a role in regulating the process.

This study provides valuable insight into the NER progression in living cells, centered around DDB2 and the ubiquitination catalyzed by the UV-DDB complex. Using Fluorescence Recovery After Photobleaching (FRAP), the authors tracked the UV-induced immobilization of various NER factors and examined the impact of the absence/presence of XPF, DDB2 and VCP segregase (that dissociated ubiquitinated DDB2 from DNA), and also the impact of ablated ubiquitination of DDB2

while retaining the ubiquitination activity of the UV-DDB E3 ligase.

A central result of the study is that blocking NER excision (using KPF KO cells or siXPG), DDB2's dissociation from lesion/proteasomal degradation (using VCP inhibitor /MG132), TFIIH verification (using siGTF2H1) or DDB2's ubiquitination (using mutant DDB2) all showed to promote accumulation/retention of DDB2 to lesions while causing an initial delay in XPC accumulation and later persistence of XPC (or XPB) at UV lesions. These results thus suggest an reciprocal relationship between lesion detection by DDB2, its ubiquitination/dissociation and the lesion handover and verification involving XPC/TFIIH: while the dissociation/degradation of DDB2 is important for the stable association of XPC and TFIIH with damaged DNA, TFIIH recruitment also must play a role in promoting the dissociation of DDB2 from damaged DNA and stable engagement of XPC on DNA.

I find the study generally convincing and important. However, there are several points that I would like to be addressed and clarified.

1. MG132 treatment didn't have any impact on DDB2 accumulation on LUD in U2OS WT cells (Supp Fig S1a,b). This indicates that proteasomal degradation is not important for DDB2 dissociation from UV lesions when NER is intact. Interestingly, even without MG132, the overall level of DDB2 didn't change after UV (Fig 2c) indicating that MG132 treatment is not likely to affect the overall level of DDB2 either. These observations make one wonder how important proteasomal degradation of DDB2 is for the NER under these WT conditions. Is it possible that the impact of Ub'd DDB2 is an anomaly that is observed only when the NER is blocked??

2. It is curious that the immobilized fraction of XPC upon UV irradiation is quite small (~10%) compared with that of DDB2 (~45%) in Figure 1. What are the expression levels of the DDB2 and XPC in those cells compared with the typical endogenous levels? Is the small immobilization of XPC due to excess amount of XPC in those cells or due to some other reason?

3. In Figure 1, can the overall XPC levels be shown for the data represented in Figure d and e (as the overall DDB2 levels are shown in Fig 1c for Figs 1a and 1b)? This would be important to show that the difference in the XPC accumulation in LUD (Figure 1d,e) is not influenced by the overall changes in the XPC level.

4. Supp Figure S1 & Fig 2. In XPF KO cells, DDB2 persists at LUD more than in WT although it gradually decreases over time (Fig 2b). Interestingly, this gradual decrease can be totally blocked by MG132 treatment according to Supp Fig S1b. However, the representative IF images shown for the data in Supp Fig S1a have very different LUD area for XPF-KO + MG132 for 40 min vs 8 h (bottom lane). Is it possible to have two cells with comparable LUD area to facilitate the comparison?

5. Also, I wonder if the MG132 treatment could preserve the overall DDB2 level that kept on disappearing in XPF KO cells (Fig 2c) and whether that could be one of the reasons that contribute to the preservation of DDB2 accumulation level in XPF KO + MG132 (Supp Fig S1b)? Can you present the overall DDB2 levels after UV for XPF KO + MG132 to the Figure 2c?

6. Figure 3c. The representative IF images for this graph is shown in Fig. S1c for 40min and 8h, but the important time point of 2 hour where there is significant difference between siCTRL and siGTF2H1 is not shown. Please show this timepoint together with others in Supp Fig. S1c.

7. Figure 3c. please show the XPC accumulation in XPF KO cells as DDB2 accumulation is shown for both U2OS WT and XPF KO cells in Figure 3b.

8. Figure 3f, g. Overall DDB2 level decrease after UV and this decrease is accelerated with siGTF2H1. However, with MG132, the decrease is stopped and also it seems DDB2 levels are low from the beginning in Fig. 3f. Why? It is still valid to focus on the lack of decrease in DDB2 over time and not consider the overall low level of DDB2?

9. (1) The results in Fig 3a, b, c show that siGTF2H1 promotes accumulation of endogenous DDB2 to LUD in WT and in XPF KO cells. (2) Fig 3d, e shows the overexpressed DDB2's ubiquitination increases with siGTF2H1 (~14% to ~19%) upon global UV irradiation. (3) The results in Fig 3f, g show that siGTF2H1 promotes gradual decrease in the overall endogenous DDB2 levels upon global irradiation (30 J/m²). (4) On the other hand, Figure 2c shows the total endogenous DDB2 protein level in U2OS WT cells over 8h after local UV irradiation (60 J/m²), determined by total nuclear fluorescence signal intensities, remains steady. Comparing (3) and (4), it seems the difference in the local and global irradiation seem to result in the difference between the steady and declining levels of DDB2. If this is the case, please provide a statement why the increase in the ubiquitinated DDB2 and the DDB2's faster disappearance which only seem to be directly observable under overexpressed DDB2 and/or global UV conditions should be relevant and important to be included in the discussion of the mechanism.

10. To make more solid point on the impact of DDB2 ubiquitination, it would be informative to show the quantification of DDB2 accumulation at LUD as done in Figs 2-5, Supp Fig S1 for the images shown in Fig 6g. It would also be important to compare this with the data shown in Figure 5b (VCPi treatment).

11. To solidify the conclusion that the impact of VCPi promoting DDB2 accumulation and diminish XPB accumulation in LUD is due to its failure in dissociating ubiquitinated DDB2 from the lesions, it would be helpful to see the data of VCPi treatment on the deltaNT and deltaNT/BP5KR mutants of DDB2.

12. Figure 4c – Is it possible to show the analogous FRAP data (time course) for XPC-GFP and GFP-XPB in the presence and absence of VCPi? If cannot be shown, please explain the reason.

13. Supp Fig 6 should include the knockdown efficiency analyses for VH10 GFP-DDB2 treated with siXPG as Supp Fig 6c shows for XP4PA XPC-GFP treated with siXPG.

(minor comments)

i. General – For all the bar graphs comparing +/- siRNA and/or inhibitors (e.g., Figs 3b, c, Figs 4b,f,g, and Figs 5 b, d, f, I), it would be helpful to have the comparison groups with clearly distinct patterns (e.g., solid for -siRNA and checkered for +siRNA) for clarity.

ii. Fig 1b – Correct DBB2 -> DDB2

iii. Fig 5g – The colors are difficult to differentiate. Please use colors that are easily distinguished from each other.

iv. Supp Fig 1 legend for (b): Isn't it "DDB2 accumulation was normalized to the nuclear background and to U2OS WT10 min after UV-C, which was set to 1.0." ?

v. Supp Fig 2 legend last line. Correct GFP-XBP -> GFP-XPB

vi. Supp Fig 4 : Remove legend for (e) and correct the one for (f) to (e) to match the figures. It seems that the Supp Fig 4e has become the main text's Figure 6f but the legend has not been corrected.

vii. Line 220: This inhibition "of" VCPi -> This inhibition "by" VCPi

viii. Lines 220-222 says "This inhibition ... was not observed after treatment with siRNA against DDB2 (Fig. 5g, h), unequivocally showing that the reduced XPC and XPB accumulation after VCPi is dependent on DDB2" but the data in Fig 5g and h are only for XPB. Please revise.

ix. Line 263. Remove "a" in 'by 266 a nm UV-C'.

Reviewer #1 (Remarks to the Author):

The dynamic hand-off of DNA damage from one recognition complex to another damage verification protein or set of proteins is an important area of research in the nucleotide excision repair pathway. In this outstanding study the authors explore the hand-off of UV damage from UV-DDB to XPC and TFIIH and the role that ubiquitylation and DDB2 degradation plays in this process. The authors have previously studied the kinetics of UV-DDB and XPC recruitment in several previous papers, but this present study uses state-of-the-art knockout and knock-in approaches provided by CRISPR/Cas9 technologies to extend this work in a highly significant manner as they can follow tagged proteins that are expressed at endogenous levels..... In total these exciting data suggest an intricate cellular timer mechanism that under normal conditions DDB2 helps recruit XPC and ubiquitylate it stabilizing at damaged sites, but if DDB2 persists at damaged sites too long, it is turned over by auto-ubiquitylation and degradation. Overall this is a timely and important contribution to the field. The experiments are well designed and executed. This already strong study would benefit from the authors consideration of the following points, which are mostly minor.

We thank the reviewer for his/her enthusiasm and highly appreciated the suggestions to improve our study, which we have addressed below.

1. In the introduction, the authors should cite the important work by Lan et al. (J Biol Chem, 287 (2012) 12036-12049 showed that UV-DDB-CUL4A-RBX binds to nucleosomes containing photoproducts, ubiquitylating H2A at Lys 119 which causes destabilization of the nucleosome.

We apologize for initially omitting this important work and have now cited it twice in our introduction, and referred to it by e.g. by writing that 'the CRL4^{DDB2} complex catalyzes monoubiquitylation of histone H2A, stimulating DDB2 extraction....'.

2. The color scheme in the FRAP traces in Figure 1a & c are difficult to follow; please use different colors.

We have adjusted the colors and line thickness of FRAP curves in Figs. 1, 4, 5 and 6 and Supplementary figures, as requested.

3. Figure 1d, what happens in a double knockdown experiment, does XPC immobilization stay low without TFIIH recruitment even in the absence of XPG? Would DDB2 remain bound?

We have performed the insightful experiment on XPC as suggested by the reviewer, which is now shown as Supplementary Fig. 1i and j. This shows that increased XPC immobilization due to XPG depletion is indeed suppressed by co-depletion of GTF2H1 (TFIIH), which supports our hypothesis that the arrival of TFIIH at damage stabilizes XPC binding. As depletion of GTF2H1 and XPG both lead to more immobilization of DDB2 (Figure 1B), we have not performed the double knockdown experiment with DDB2. Our IF data shown in Fig. 3a and b indicates that there is increased accumulation of DDB2 in XPG KO cells (equivalent to XPG depletion) when TFIIH is depleted.

4. It would be good to show control westerns indicating the level of knockdown achieved for each siRNA used. For example, GTF2H1 siRNA KD is shown in Figure 1, but the western is not shown until Figure 3. It does not look like XPG or DDB2 levels were ever shown after KD with siRNA. Also are there accompanying westerns for DDB2 and a control protein for the experiments described in 4?

We apologize for not showing this in more logical order. In the original Supplementary Fig. 6, to which we referred in the Methods, we showed knockdown efficiencies of siRNAs used. We have now moved this figure and made it Supplementary Fig. 1, to which we immediately refer in the beginning of the results.

5. Lines 296-297 the authors state that XPC searches with “random motion”. In fact, two single molecule studies cited by the authors show anomalous diffusion of Rad4 and XPC around CPD. This anomalous diffusion could provide dynamic recognition of helping to recruit both UV-DDB and TFIIH to the vicinity of the lesion site without binding directly to the damaged site, which might interfere with subsequent repair events. Thus, at the scale of single molecules binding events might be in fact transient and dynamic and the FRAP experiments described here which are bulk analysis and not able to provide such spatial and temporal resolution. The authors actually revisit this issue again on lines 321-323.

We thank the reviewer for his/her insightful comment. In our description of a model for the dynamic search mechanism of XPC, which is based on multiple single molecule studies from different labs, we have followed the ‘unified’ model as presented by Mu et al 2018 (DNA repair; PMID 30174301). By no means do we mean to indicate that our FRAP results have a similar spatial and temporal resolution. We describe the molecular model of XPC damage search (based on single molecule studies) because it fits very well with, and provides an explanation for, our FRAP results. We agree that TFIIH could be recruited before any stable binding to damage occurs, which we also suggest. Because it is unclear to us whether the reviewer requested any change to our description in the text and what exactly should be changed, we have not made any changes in the text based on this comment.

6. To this end, perhaps proximity ligation assays could be used to show direct interaction of DDB2-XPC or XPC-TFIIH, but not DDB2-TFIIH.

We agree that proximity ligation assays would be a great way, if successful, to show whether direct interactions occur or not. For this reason, we have performed these assays of which we present the data below. We can clearly detect a UV-induced increase in foci for XPC-TFIIH(GTF2H1) and TFIIH(GTF2H1)-XPA. We indeed did not find any UV-induced interaction between DDB2 and TFIIH, which we tested with two different antibody combinations (DDB2ms-GTF2H1rb and DDB2rb-GTF2H1ms). However, also two positive controls, i.e. DDB2 with Cul4A and with DDB1, did not show any clear foci. Because of this, we cannot be sure that the DDB2 antibodies used, which to our knowledge are the best for western and IF currently available on the market, work well in the PLA assay to confirm or disprove interactions between DDB2 and XPC or TFIIH. For this reason, we have chosen not to include these results in our

manuscript.

7. The discussion provides an exciting model that is consistent with their data. However, it suffers in places, by preferential citation of some, but not all of the literature – this needs correcting. For example: - Lines 324-325, The model that UV-DDB kinks the DNA duplex while supported by the Thoma structure (ref 15) is not supported by the Yeh structure (Proc Natl Acad Sci U S A, 109 (2012) E2737-2746) which showed little or no kink.

We thank the reviewer for these helpful insights and apologize if our discussion appeared to preferentially cite some work over others, which is not our intent. We have now included the Yeh 2012 paper (in the introduction and discussion) and removed the notion that UV-DDB kinks the DNA.

- Lines 334-336, Simultaneous binding of XPC and UV-DDB was modeled by Thoma's group in their Scrima Cell paper (ref 15), see supplemental figure S7 where a co-complex is modeled.

We are aware of the modeling of simultaneous binding of DDB2 and XPC, which is why we cite this work to support the idea that DDB2 and XPC should temporarily coexist together, mentioning that XPC 'uses separate domains to bind to DNA adjacent and opposite of the lesion'.

- Line 344-345, That NER is inhibited by addition of DDB2 is inconsistent with work by Matsunaga and coworkers, who using a fully reconstituted NER system consisting of seven purified proteins and 136 base pair (bp) DNA substrates containing either a CPD or 6-4PP lesion, demonstrated that UV-DDB could stimulate excision of CPD by 5-17-fold in a reaction containing 571 fmol of UV-DDB, but only displayed a ~2 fold stimulation for 6-4PP even when used at a significantly reduced amount (0.57 – 5.7 fmol) J Biol Chem, 277 (2002) 1637-1640.). It is also interesting to note that UV-DDB was found by in the crystal structure and by EM and AFM to form dimers of dimers in the Yeh et al paper cited above. Perhaps these dimers of dimers of UV-DDB, which bind very tightly to the DNA are inhibitor to repair and help explain the lack of congruence between the JBC 2002 paper by the Matsunaga group and the Cell 121, 387–400 (2005) by Sugawara et al.

We thank the reviewer for these considerations. We do not merely claim that 'NER is inhibited by addition of DDB2' but cite the work of Sugawara that shows that '*in vitro* reconstituted NER' is inhibited by DDB2 'in the absence of ubiquitylation factors that mediate its release' (this is how we formulated this). This is an observation made by others and not by us, so we do not think that we have to justify their results if they contradict the work of Matsunaga et al 2002. Importantly, the *in vitro* studies of Matsunaga and of Sugawara use different experimental designs (including different concentrations of repair template and proteins), so it is difficult to compare these directly. However, also in Matsunaga 2002 with excess amounts of UV-DDB, there is inhibition of 6-4PP repair. Therefore, we now have revised and better specified our statement and write that 'the addition of excess DDB2 in the absence of ubiquitylation factors' inhibits NER 'of 6-4PPs' *in vitro*, citing both the work of Sugawara and of Matsunaga. Furthermore, we agree that UV-DDB dimerization on the DNA could be inhibitory to repair and have now included this idea in our discussion, citing the Yeh 2012 paper and also more recent work on DDB2 (Jang et al 2019).

8. The working model shown in Figure 7 is helpful, but not entirely consistent with their own data. For example, the authors state on lines 388-389, "Dissociated ubiquitylated DDB2 is targeted for

proteasomal degradation, which decreases its effective concentration.” This is only true when XPF is knocked out or XPG is knocked down and as stated on lines 182-184, “Our observations suggest that the recruitment of TFIIH promotes the stable binding of XPC to damaged DNA and the dissociation of DDB2, thereby preventing excessive DDB2 auto-ubiquitylation and degradation.” Thus, the authors should include a caveat to their figure that only during unbalanced NER (i.e. decrease TFIIH recruitment or loss of incision or high damaged levels) does UV-DDB become highly ubiquitylated to such a degree that it is degraded in an appreciable manner. The UV-DDB direct ubiquitylation of XPC and its auto-ubiquitylation is three-fold, helping to recruit XPC at damaged sites and allow UV-DDB dissociation and subsequent TFIIH recruitment. This study and their working model thus helps resolve a long-standing question in the field as to why a cell might kill of its first responder, UV-DDB, through protein degradation. Furthermore, one might imagine a DUB that helps deubiquitylate DDB2 after its dissociation and allowing DDB2 to undergo another round of damage recognition and XPC loading.

We thank the reviewer for this suggestion. However, we think that ubiquitylated DDB2 is targeted for proteasomal degradation also in wild type cells, but to a much lesser extent (=less molecules) than in NER deficient cells, as is also shown in Fig. 3f and g. This degradation is enhanced in cells that lack XPF or XPG, due to the continuous rebinding of DDB2 molecules (and thus ubiquitylation of more DDB2 molecules) to unrepaired lesions. Thus, our model predicts that if more DDB2 molecules bind (i.e. in situation of higher UV dose, deficient NER but also when there are more DDB2 molecules present), more ubiquitylation and degradation take place and can thus be measured. To make this more clear, as the reviewer suggests, we have now included in the text of the discussion, as well as indicated in Fig. 7, that when NER is deficient this targeting for degradation is enhanced. Furthermore, we agree that it would very well be possible that a DUB is involved in regulating DDB2 concentrations, but as this is (at this stage) still speculative and not based on our current data, we have not included this idea in our manuscript.

9. While this might be beyond the scope of the present work, where is ubiquitylated DDB2 being degraded? Within the nucleus or in the cytoplasm? Is it known if the nucleus has active 26S proteasomes. Some speculation in the discussion about this matter would be helpful to the reader.

The reviewer raises an interesting question. Proteasomal components are indeed found in the nucleus, but, to the best of our knowledge, it is still debated whether these are proteolytically active or not. And even if they are actively degrading, this still does not necessarily mean that any nuclear protein is also degraded in the nucleus. As for DDB2, its ubiquitylated form was found localized in nuclei and not in the cytoplasm (PMID 12034848), but we are unsure whether this necessarily means that it is also degraded in the nucleus. The answer to this question would require substantial dedicated research. Because we think this matter is still very speculative and beyond the scope of our work, as the reviewer also indicates, and because it does not impact our results or conclusions directly, we have chosen not to include this in our discussion.

Reviewer #2 (Remarks to the Author):

In this manuscript, the authors focus on mechanistic regulation of the DNA damage recognition process in mammalian global genomic nucleotide excision repair (GG-NER). For efficient GG-NER of UV-induced DNA damage, coordinated actions of multiple protein factors, including DDB1-DDB2 (UV-DDB), XPC, TFIIH, etc., are crucial. Regulation of this process involves protein-protein and protein-DNA interactions as well as post-translational modifications such as ubiquitination, which has not yet been fully understood.....

Overall, the manuscript is well written, providing data from experiments that are appropriately designed and performed with high technical quality. The conclusions they made are mostly convincing and supported by the presented results. This reviewer has only minor comments and suggestions, which the authors could consider for further improvement.

We thank the reviewer for his/her positive evaluation of our work and highly appreciate the suggestions to improve the manuscript, which we address below.

1. Some FRAP data are difficult to see, because of overlapping lines (especially Fig. 5g). The authors could consider to change colors of lines and/or separate a graph into multiple ones.

We have adjusted the colors and line thickness of FRAP curves in Fig. 1, 4, and 6 and Supplementary Figures and separated the graphs in Fig. 5g.

2. Fig. 1b: Title of the graph should be corrected (DDB2).

We have corrected the typo in this title.

3. Fig. 3d: This data would be even more convincing, if the authors could include results with control cells lacking GFP-DDB2 expression and show that the UV-induced FK2 signals are not detected.

The reviewer is right that this data would be more convincing with this important control experiment. In the revised version of the manuscript, we have now included this control as Supplementary Fig. 2f, which shows that indeed UV-induced FK2 signals are only detected with GFP-DDB2 immunoprecipitation and not with GFP only.

4. Fig. 6c: This reviewer expects that, after UV irradiation, part of CUL4A undergoes conjugation to NEDD8 and shows a band shift. In addition, amounts of DDB1 in IP fractions do not seem to correlate with GFP signals (especially, in the right most lanes). Does this suggest instability of the complex?

We thank the reviewer for his/her considerations and agree that a shifted CUL4A band is expected. Indeed, when CUL4A is neddylated it shows a band shift. However, this is not visible on our blot as the vast majority of CUL4A bound to UV-DDB in the type of IP shown in Fig. 6 is not bound to chromatin, as the purpose of this IP was not to isolate chromatin-bound CUL4A-DDB2 complexes but to test if the CUL4A-DDB2 complex is formed with the GFP-tagged DDB2 versions. If we perform different IPs to focus on the neddylation of CUL4A, we can indeed observe that a minor fraction is neddylated after UV as shown in the example GFP-DDB2 IP here provided. This work, however, is part of another manuscript that is currently in

preparation. For this reason, we prefer (and kindly ask) not to focus on this minor matter.

We performed these IP experiments to verify that the CRL4-DDB2 complex is formed, which is indeed suggested by the co-IP of CUL4A, DDB1 and CSN5. Although the DDB1 signals in the IP fraction shown may appear to vary a bit, this is not something we repeatedly observed. For instance, below is another GFP-DDB2 IP experiment shown in which we stained for DDB1 (which does not stain easily on western blot) which does not show this variation. Therefore, we do not think these results necessarily point to instability of the complex, which is why we have not emphasized this in the manuscript.

5. Suppl. fig. 2a: This Western blot shows doublet bands of DDB2, which seems different from other DDB2 blots. Could the authors assign each of these species?

We indeed sometimes observe multiple bands with the used antibody in whole cell extracts. In this western, the lower band is DDB2, as this bands disappears with siRNA (as depicted e.g. in Supplementary Fig. 1e). We do not always observe the second, likely nonspecific higher band when blotting DDB2 and therefore do not know what this second band is. To make the figure more clear, we have repeated the western blot to show DDB2 KO (which is Supplementary Fig. 3a in the revised manuscript) using new cell extracts, which now only shows one DDB2 band in WT that disappears in the KO.

6. Suppl. fig. 4: According to the figure legend, panel e seems missing, and the graph labeled "e" should be panel f.

We thank the reviewer for noticing this mistake, which we have now corrected (in the legend of Supplementary Fig. 5 of the revised manuscript).

Reviewer #3 (Remarks to the Author):

This manuscript by Ribeiro-Silva describes cellular fluorescence imaging studies that shed light on the mechanism of progression in NER during damage recognition and verification..... These results thus suggest an reciprocal relationship between lesion detection by DDB2, its ubiquitination/dissociation and the lesion handover and verification involving XPC/TFIIH: while the dissociation/degradation of DDB2 is important for the stable association of XPC and TFIIH with damaged DNA, TFIIH recruitment also must play a role in promoting the dissociation of DDB2 from damaged DNA and stable engagement of XPC on DNA.

I find the study generally convincing and important. However, there are several points that I would like to be addressed and clarified.

We thank the reviewer for his/her enthusiasm for our study and the raised points to improve the quality of the manuscript, which we address below.

1. MG132 treatment didn't have any impact on DDB2 accumulation on LUD in U2OS WT cells (Supp Fig S1a,b). This indicates that proteasomal degradation is not important for DDB2 dissociation from UV lesions when NER is intact. Interestingly, even without MG132, the overall level of DDB2 didn't change after UV (Fig 2c) indicating that MG132 treatment is not likely to affect the overall level of DDB2 either. These observations make one wonder how important proteasomal degradation of DDB2 is for the NER under these WT conditions. Is it possible that the impact of Ub'd DDB2 is an anomaly that is observed only when the NER is blocked??

We thank the reviewer for this suggestion. We indeed observe that DDB2 proteasomal degradation is uncoupled from its dissociation, as we also mention in the discussion that *'the degradation of DDB2..... is separate from its extraction and dissociation from DNA'*. However, overall levels of DDB2 do change in NER proficient cells after UV, albeit to a much lesser extent than in NER deficient cells, as shown in Fig. 3f and g (which also shows this is inhibited by MG132). In the absence of NER, simply more DDB2 molecules bind to lesions and therefore more molecules are ubiquitylated and degraded, which is why with limited, localized UV damage as in Fig. 2c, the DDB2 degradation is only measurable in NER deficient cells and not in NER proficient cells. To make more clear that this process, in our opinion, is enhanced (rather than an anomaly) in NER deficient cells, we have now indicated this specifically in the model shown in Fig. 7 and in the text describing this figure.

2. It is curious that the immobilized fraction of XPC upon UV irradiation is quite small (~10%) compared with that of DDB2 (~45%) in Figure 1. What are the expression levels of the DDB2 and XPC in those cells compared with the typical endogenous levels? Is the small immobilization of XPC due to excess amount of XPC in those cells or due to some other reason?

The reviewer is correct in noticing the difference between DDB2 and XPC immobilization in FRAP. For our FRAP experiments, we chose to use established and published VH10 GFP-DDB2 expressing cells (PMID 23045548). In this cell line, GFP-DDB2 is overexpressed and this leads to a relatively large immobilized fraction, because DDB2 has strong affinity for DNA damage and its dissociation is promoted (as we show in our manuscript) by stable XPC and TFIIH binding (whose levels are not increased). XPC behaves differently because its overexpression leads to relatively smaller immobile fractions. XPC-GFP is slightly overexpressed in the established and published cell line that we chose to use for our FRAP analyses (PMID 18682493; 30287812). For instance, Figure 4C of Steurer et al 2019 (PMID 30698791) shows that GFP-XPC expressed at lower endogenous levels, in GFP-XPC knock-in cells, has a larger

(relative) immobile fraction. In the revised manuscript, we have included a FRAP experiment in the same GFP-XPC knock-in cells (Fig. 5i) in answer to point 12 below, which also shows this larger immobilization. Importantly, because our FRAPs are not performed on GFP-tagged proteins expressed at endogenous levels, we do not perform any absolute quantitative comparisons between the DDB2 and XPC immobilize fractions but only quantitatively compare results obtained with different siRNAs in the same cells. Also, for that reason, we substantiated the FRAP experiments with IF experiments studying endogenous proteins.

3. In Figure 1, can the overall XPC levels be shown for the data represented in Figure d and e (as the overall DDB2 levels are shown in Fig 1c for Figs 1a and 1b)? This would be important to show that the difference in the XPC accumulation in LUD (Figure 1d,e) is not influenced by the overall changes in the XPC level.

We assume that the reviewer means Fig. 2d-e and we agree that it is important to show that the overall XPC levels do not change. We now show this in the new Supplementary Fig. 2d.

4. Supp Figure S1 & Fig 2. In XPF KO cells, DDB2 persists at LUD more than in WT although it gradually decreases over time (Fig 2b). Interestingly, this gradual decrease can be totally blocked by MG132 treatment according to Supp Fig S1b. However, the representative IF images shown for the data in Supp Fig S1a have very different LUD area for XPF-KO + MG132 for 40 min vs 8 h (bottom lane). Is it possible to have two cells with comparable LUD area to facilitate the comparison?

We thank the reviewer for his/her suggestion to include a better representative image and have adjusted this figure according to the reviewer's request, which is now shown in Supplementary Fig. 2a.

5. Also, I wonder if the MG132 treatment could preserve the overall DDB2 level that kept on disappearing in XPF KO cells (Fig 2c) and whether that could be one of the reasons that contribute to the preservation of DDB2 accumulation level in XPF KO + MG132 (Supp Fig S1b)? Can you present the overall DDB2 levels after UV for XPF KO + MG132 to the Figure 2c?

We indeed think that the preservation of DDB2 accumulation in XPF KO cells is because MG132 preserves overall DDB2 levels (as also discussed in the text), allowing continuous targeting of DDB2 to unrepaired DNA damage sites. We now show the overall DDB2 levels in these cells in the new Supplementary Fig. 2c.

6. Figure 3c. The representative IF images for this graph is shown in Fig. S1c for 40min and 8h, but the important time point of 2 hour where there is significant difference between siCTRL and siGTF2H1 is not shown. Please show this timepoint together with others in Supp Fig. S1c.

We agree that it is good to include representative images of this time point. As the reviewer requests, we have now included the 2 h time point in the IF images that are shown in the (revised) Supplementary Fig. 2e.

7. Figure 3c. please show the XPC accumulation in XPF KO cells as DDB2 accumulation is shown for both U2OS WT and XPF KO cells in Figure 3b.

As the reviewer requests, in the revised manuscript, we now show XPC accumulation with and without siGTF2H1 in the XPF KO cells as well (Fig. 3c and Supplementary Fig. 2e.).

8. Figure 3f, g. Overall DDB2 level decrease after UV and this decrease is accelerated with siGTF2H1. However, with MG132, the decrease is stopped and also it seems DDB2 levels are low from the beginning in Fig. 3f. Why? It is still valid to focus on the lack of decrease in DDB2 over time and not consider the overall low level of DDB2?

We understand that from Fig. 3f it might appear that with MG132 the levels of DDB2 are low from the beginning. However, the depicted images are from different immunoblots that were independently stained with antibodies and scanned for fluorescence and therefore cannot be compared in a quantitative manner with each other. Supplementary Fig. 4a depicts a western blot in which DDB2 from cells without and with MG132 is loaded on the same gel, from which it is clear that MG132 does not influence initial DDB2 levels.

9. (1) The results in Fig 3a, b, c show that siGTF2H1 promotes accumulation of endogenous DDB2 to LUD in WT and in XPF KO cells. (2) Fig 3d, e shows the overexpressed DDB2's ubiquitination increases with siGTF2H1 (~14% to ~19%) upon global UV irradiation. (3) The results in Fig 3f, g show that siGTF2H1 promotes gradual decrease in the overall endogenous DDB2 levels upon global irradiation (30 J/m²). (4) On the other hand, Figure 2c shows the total endogenous DDB2 protein level in U2OS WT cells over 8h after local UV irradiation (60 J/m²), determined by total nuclear fluorescence signal intensities, remains steady. Comparing (3) and (4), it seems the difference in the local and global irradiation seem to result in the difference between the steady and declining levels of DDB2. If this is the case, please provide a statement why the increase in the ubiquitinated DDB2 and the DDB2's faster disappearance which only seem to be directly observable under overexpressed DDB2 and/or global UV conditions should be relevant and important to be included in the discussion of the mechanism.

The reviewer correctly notices that DDB2 ubiquitylation and degradation are more easily detectable when more UV damage is generated (global versus local) or when more DDB2 molecules are present. However, 'DDB2's faster disappearance' is also observable (though less prominent) by measuring endogenous protein levels after local UV damage, as is evident from DDB2 levels after local UV damage in the XPF KO cells (Fig. 2c). Our model predicts that if more DDB2 molecules bind, also more DDB2 molecules are ubiquitylated and (consequently) degraded. That is why, if a certain amount of local UV damage is applied (60 J/m² through an 8 μm pore), we only clearly see a reduction in endogenous DDB2 levels in XPF KO cells (as in these cells there is continuous, i.e. more, binding of DDB2; Fig. 2c). But if we apply much higher levels of UV damage (30 J/m² global UV; and thus more DDB2 molecules bind damage), we can also measure degradation in wild type cells (which is enhanced with siGTF2H1; Fig. 3f, g). To make this more clear, we have now indicated this in our model in Fig. 7 and also added the following sentence in our discussion of this model: 'When more DDB2 molecules bind to lesions, e.g. in case of deficient NER or higher DNA damage load, more molecules are ubiquitylated and thus proteasomal degradation is enhanced'.

10. To make more solid point on the impact of DDB2 ubiquitination, it would be informative to show the quantification of DDB2 accumulation at LUD as done in Figs 2-5, Supp Fig S1 for the images shown in Fig 6g. It would also be important to compare this with the data shown in Figure 5b (VCPi treatment).

As the reviewer requested, we have quantified the DDB2 accumulation at LUD in the cells of the experiments shown in Fig. 6g and h. Because XPC and CPD are visualized with rabbit and mouse antibodies, we could not use additional antibodies to clearly visualize DDB2 in these IFs. The IF procedure tends to diminish the GFP-DDB2 fluorescence, due to the NaOH treatment which is required

to visualize CPDs but which quenches GFP signal. As in the slides of the 40 min time point the GFP signal was most uniform, we only quantified and show the 40 min time point in the revised manuscript (Fig. 6g, h and Supplementary Fig. 5e).

11. To solidify the conclusion that the impact of VCPi promoting DDB2 accumulation and diminish XPB accumulation in LUD is due to its failure in dissociating ubiquitinated DDB2 from the lesions, it would be helpful to see the data of VCPi treatment on the deltaNT and deltaNT/BP5KR mutants of DDB2.

As described in the manuscript, our intent in making the DDB2 mutants was to circumvent the use of VCPi, which also diminishes XPC ubiquitylation, to show that DDB2 retention at damage (without affecting XPC ubiquitylation) inhibits XPC and XPB recruitment. However, as the reviewer requests we have tested the impact of VCPi on both DDB2 mutants and find that only the deltaNT/BP5KR is completely unaffected by VCPi (Supplementary Fig. 5f). This is likely because this mutant lacks 5 lysine residues that can still be ubiquitylated in the deltaNT mutant (see e.g. PMID 25628365). Interestingly, we have tested the impact of VCPi on XPC and XPB recruitment in cells expressing deltaNT/BP5KR GFP-DDB2 and find that this further reduces XPC and XPB recruitment. These new experiments are now shown in Supplementary Fig. 5g and h and further support the idea that both DDB2 dissociation and XPC ubiquitylation promote the stable association of XPC and TFIIH to DNA damage.

12. Figure 4c – Is it possible to show the analogous FRAP data (time course) for XPC-GFP and GFP-XPB in the presence and absence of VCPi? If cannot be shown, please explain the reason.

We agree that it would be interesting to show the analogous FRAP data. For XPB, this data is (already) shown in Fig. 5g,h. For XPC, we have performed the requested FRAP experiments, in DDB2 proficient but also in DDB2 KO cells, which are now shown in Fig. 5i and Supplementary Fig. 3g. These new data further support the IF experiments shown in Fig. 5c, d.

13. Supp Fig 6 should include the knockdown efficiency analyses for VH10 GFP-DDB2 treated with siXPG as Supp Fig 6c shows for XP4PA XPC-GFP treated with siXPG.

As the reviewer requests, we include in the revised manuscript the knockdown efficiency analysis for VH10 GFP-DDB2 with siXPG. The figures have been moved and are now shown as Supplementary Fig. 1.

(minor comments)

i. General – For all the bar graphs comparing +/- siRNA and/or inhibitors (e.g., Figs 3b, c, Figs 4b,f,g , and Figs 5 b, d, f, I), it would be helpful to have the comparison groups with clearly distinct patterns (e.g., solid for -siRNA and checkered for +siRNA) for clarity.

We understand the reviewer's request and have also ourselves tried different ways of representing the data. However, due to the policy of the journal to show all data points in the graph, we have chosen to depict our graphs as all data points overlaid with the bars. Because of this, we do not think that adding another layer of lines/dots (in the form of checkered bars) will greatly improve clarity.

ii. Fig 1b – Correct DBB2 -> DDB2

We thank the reviewer for thoroughly reading the manuscript and pointing to several textual inaccuracies. We have corrected the this typo, as well as others pointed out below.

iii. Fig 5g – The colors are difficult to differentiate. Please use colors that are easily distinguished from each other.

We have changed the colors and (in answer to reviewer 2) split the graph for clarity.

iv. Supp Fig 1 legend for (b): Isn't it "DDB2 accumulation was normalized to the nuclear background and to U2OS WT10 min after UV-C, which was set to 1.0." ?

We thank the reviewer for noticing this mistake, which we have corrected.

v. Supp Fig 2 legend last line. Correct GFP-XBP -> GFP-XPB

We have corrected the mistake.

vi. Supp Fig 4 : Remove legend for (e) and correct the one for (f) to (e) to match the figures. It seems that the Supp Fig 4e has become the main text's Figure 6f but the legend has not been corrected.

We thank the reviewer for noticing this mistake, which we have corrected.

vii. Line 220: This inhibition "of" VCPi -> This inhibition "by" VCPi

We have corrected the mistake.

viii. Lines 220-222 says "This inhibition ... was not observed after treatment with siRNA against DDB2 (Fig. 5g, h), unequivocally showing that the reduced XPC and XPB accumulation after VCPi is dependent on DDB2" but the data in Fig 5g and h are only for XPB. Please revise.

With the newly added experiments (Fig. 5i and Supplementary Fig. 3g), we now also show for XPC that the VCPi effect depends on DDB2 (which was also shown with the IFs in Fig 5c,d). Therefore, we think that the statement is now more accurate.

ix. Line 263. Remove "a" in 'by 266 a nm UV-C'.

We have corrected the mistake.

Reviewers' Comments:

Reviewer #1:

Remarks to the Author:

This revised manuscript has fully dealt with the concerns raised in the previous review. The authors, in many cases, have provided additional experiment support to address the concerns, and when not feasible have explained why the experiments are beyond the scope of the present work. Overall this is an important study that addresses the nature of DDB2 ubiquitination and degradation after UV-C. Furthermore the authors nicely show that TFIIH arrival to the damaged site helps dissociate DDB2 and stabilize XPC. Overall this is an important and timely study that will have a lasting impact on the field.

Reviewer #2:

Remarks to the Author:

The authors addressed this reviewer's concerns in a mostly appropriate manner. He/she still feels that the blots in Fig. 6c would better be replaced, if possible, to avoid confusion. To make the authors' conclusions very convincing, it is not a minor point that this specific mutant is defective only in ubiquitination, but not in other functions.

Reviewer #3:

Remarks to the Author:

This manuscript by Ribeiro-Silva describes cellular fluorescence imaging studies that shed light on the mechanism of progression in NER during damage recognition and verification. While the step-wise biochemical processes of global genome NER have been known for a while, our understanding of how the seemingly sequential steps are coordinated in cells has been relatively poor. There are not only multiple protein factors working within chromatinized DNA environment but also multiple distinct posttranslational modifications for the proteins factors that play a role in regulating the process.

This study provides valuable insight into the NER progression in living cells, centered around DDB2 and the ubiquitination catalyzed by the UV-DDB complex. Using Fluorescence Recovery After Photobleaching (FRAP), the authors tracked the UV-induced immobilization of various NER factors and examined the impact of the absence/presence of XPG, XPF, DDB2 and VCP segregase (that dissociated ubiquitinated DDB2 from DNA), and also the impact of ablated ubiquitination of DDB2 while retaining the ubiquitination activity of the UV-DDB E3 ligase.

A central result of the study is that blocking NER excision (using XPF KO cells or siXPG), DDB2's dissociation from lesion/proteasomal degradation (using VCP inhibitor /MG132), TFIIH verification (using siGTF2H1) or DDB2's ubiquitination (using mutant DDB2) all showed to promote accumulation/retention of DDB2 to lesions while causing an initial delay in XPC accumulation and later persistence of XPC (or XPB) at UV lesions. These results thus suggest an reciprocal relationship between lesion detection by DDB2, its ubiquitination/dissociation and the lesion handover and verification involving XPC/TFIIH: while the dissociation/degradation of DDB2 is important for the stable association of XPC and TFIIH with damaged DNA, TFIIH recruitment also must play a role in promoting the dissociation of DDB2 from damaged DNA and stable engagement of XPC on DNA.

The revised manuscript addresses all my concerns and I believe the manuscript merits publication in the journal.

Response to reviewers' comments

Reviewer #1 (Remarks to the Author):

This revised manuscript has fully dealt with the concerns raised in the previous review. The authors, in many cases, have provided additional experiment support to address the concerns, and when not feasible have explained why the experiments are beyond the scope of the present work. Overall this is an important study that addresses the nature of DDB2 ubiquitination and degradation after UV-C. Furthermore the authors nicely show that TFIIH arrival to the damaged site helps dissociate DDB2 and stabilize XPC. Overall this is an important and timely study that will have a lasting impact on the field.

We thank the reviewer again for his/her constructive criticism and are happy to read that we have satisfactorily dealt with the raised concerns.

Reviewer #2 (Remarks to the Author):

The authors addressed this reviewer's concerns in a mostly appropriate manner. He/she still feels that the blots in Fig. 6c would better be replaced, if possible, to avoid confusion. To make the authors' conclusions very convincing, it is not a minor point that this specific mutant is defective only in ubiquitination, but not in other functions.

We thank the reviewer again for his/her constructive criticism and for accepting our responses to the raised concerns. We understand the concern with the blot in Fig 6c, but this blot is only intended to show (together with the blot in Supplementary fig 5c) that the CRL4-DDB2 complex is formed and functional. Therefore, we do not think this will lead to much confusion.

Reviewer #3 (Remarks to the Author):

This manuscript by Ribeiro-Silva describes cellular fluorescence imaging studies that shed light on the mechanism of progression in NER during damage recognition and verification. While the step-wise biochemical processes of global genome NER have been known for a while, our understanding of how the seemingly sequential steps are coordinated in cells has been relatively poor. There are not only multiple protein factors working within chromatinized DNA environment but also multiple distinct posttranslational modifications for the proteins factors that play a role in regulating the process. This study provides valuable insight into the NER progression in living cells, centered around DDB2 and the ubiquitination catalyzed by the UV-DDB complex. Using Fluorescence Recovery After Photobleaching (FRAP), the authors tracked the UV-induced immobilization of various NER factors and examined the impact of the absence/presence of XPG, XPF, DDB2 and VCP segregase (that dissociated ubiquitinated DDB2 from DNA), and also the impact of ablated ubiquitination of DDB2 while retaining the ubiquitination activity of the UV-DDB E3 ligase.

A central result of the study is that blocking NER excision (using XPF KO cells or siXPG), DDB2's dissociation from lesion/proteasomal degradation (using VCP inhibitor /MG132), TFIIH verification (using siGTF2H1) or DDB2's ubiquitination (using mutant DDB2) all showed to promote accumulation/retention of DDB2 to lesions while causing an initial delay in XPC accumulation and later persistence of XPC (or XPB) at UV lesions. These results thus suggest an reciprocal relationship between lesion detection by DDB2, its ubiquitination/dissociation and the lesion handover and verification involving XPC/TFIIH: while the dissociation/degradation of DDB2 is important for the stable association of XPC and TFIIH with damaged DNA, TFIIH recruitment also must play a role in promoting the dissociation of DDB2 from damaged DNA and stable engagement of XPC on DNA.

The revised manuscript addresses all my concerns and I believe the manuscript merits publication in the journal.

We thank the reviewer again for his/her constructive criticism and positive evaluation of the revised manuscript.